# *C. elegans* XMAP215/ZYG-9 and TACC/TAC-1 act at multiple times during oocyte meiotic spindle assembly and promote both spindle pole coalescence and stability

**Austin M. Harvey**, **Chien-Hui Chuang, Eisuke Sumiyoshi, Bruce Bowerman** *

Institute of Molecular Biology, University of Oregon, Eugene, Oregon, United States of America

* bowerman@uoregon.edu

**Data Availability Statement:** All relevant data are within the manuscript and its Supporting Information files.

## Abstract

The conserved two-component XMAP215/TACC modulator of microtubule stability is required in multiple animal phyla for acentrosomal spindle assembly during oocyte meiotic cell division. In *C. elegans*, *XMAP215/zyg-9* and *TACC/tac-1* mutant oocytes exhibit multiple and indistinguishable oocyte spindle assembly defects beginning early in meiosis I. To determine if these defects represent one or more early requirements with additional later and indirect consequences, or multiple temporally distinct and more direct requirements, we have used live cell imaging and fast-acting temperature-sensitive *zyg-9* and *tac-1* alleles to dissect their requirements at high temporal resolution. Temperature upshift and downshift experiments indicate that the ZYG-9/TAC-1 complex has multiple temporally distinct and separable requirements throughout oocyte meiotic cell division. First, we show that during prometaphase ZYG-9 and TAC-1 promote the coalescence of early pole foci into a bipolar structure, stabilizing pole foci as they grow and limiting their growth rate, with these requirements being independent of an earlier defect in microtubule organization that occurs upon nuclear envelope breakdown. Second, during metaphase, ZYG-9 and TAC-1 maintain spindle bipolarity by suppressing ectopic pole formation. Third, we show that ZYG-9 and TAC-1 also are required for spindle assembly during meiosis II, independently of their meiosis I requirements. The metaphase pole stability requirement appears to be important for maintaining chromosome congression, and we discuss how negative regulation of microtubule stability by ZYG-9/TAC-1 during oocyte meiotic cell division might account for the observed defects in spindle pole coalescence and stability.

## Author summary

When most animal cells divide, large multiprotein complexes, called centrosomes, nucleate and organize protein filaments, called microtubules, into a dynamic bipolar structure called the spindle that equally partitions the duplicated genome between two daughter cells. However, female oocytes lack centrosomes but still assemble bipolar spindles that

**Funding:** This work was supported by funding from the National Institutes of Health (www.nih.gov) R35GM131749 (AMH, C-HC, ES, and BB) and T32GM007413 (AMH). The funders had no role in study design, data collection and analysis, decision to publish, or preparation of the manuscript.

**Competing interests:** No competing interests exist.

separate chromosomes. Using the nematode *C. elegans* as a model system, and taking advantage of fast-acting temperature-sensitive mutations that rapidly inactivate or reactivate proteins upon temperature upshifts or downshifts, respectively, we show that a complex of two regulators of microtubule stability, called ZYG-9 and TAC-1, has multiple and separable requirements during acentrosomal oocyte spindle assembly. These requirements include promoting the coalescence of early pole foci into a bipolar structure, and the subsequent maintenance of pole stability, both of which are essential for proper chromosome separation. Furthermore, oocytes undergo two consecutive cell divisions to produce an egg with a single copy of the genome, and we show that ZYG-9 and TAC-1 are required for pole coalescence during both the first and second of these meiotic cell divisions. Our findings provide a high resolution view of the distinct and separable temporal requirements for these widely conserved regulators of microtubule stability during acentrosomal oocyte spindle assembly.

## Introduction

Oocyte meiosis I and II are sequential, highly asymmetric cell divisions that reduce a duplicated genome to single copy, producing a haploid egg. Meiotic cell division requires a microtubule-based spindle apparatus to partition the genome and, in contrast to mitotic cells [1], the oocytes of most animal models lack centrosomes but nevertheless nucleate and organize microtubules into functional bipolar spindles [2–5]. Faithful transmission of the genome is essential, but the molecular mechanisms underlying acentrosomal oocyte meiotic spindle assembly and function *in vivo* remain poorly understood.

In *Caenorhabditis elegans*, oocyte spindle assembly occurs through a sequence of stages defined by visible changes in microtubule dynamics [6,7]. First is nuclear envelope breakdown (NEBD) and the entry of tubulin into the nucleus. Microtubule bundles then appear beneath the disassembling nuclear lamina, forming a peripheral and roughly spherical microtubule network at the cage stage. Next, during the multipolar stage, microtubules increase in amount and assemble into a network with numerous nascent pole foci marked with the pole protein ASPM-1 [8]. Over time these pole foci coalesce, forming a bipolar spindle by metaphase, with homologous chromosome pairs aligned between the poles.

Following these stages in bipolar spindle assembly, the oocyte transitions into anaphase, when the spindle undergoes extensive morphological changes during chromosome separation [9,10]. First, the spindle shortens and the poles broaden, as the spindle rotates to orient perpendicularly to the cortex [11], with the two separating chromosome sets each moving slightly towards the nearest pole during anaphase A. Subsequently, the spindle poles largely disassemble while parallel arrays of microtubules appear and elongate between the separating chromosome sets during anaphase B [12–14], with half of the genome extruded into the first polar body upon the completion of meiosis I. While this description provides a platform for further investigation, when and how the proteins required for oocyte spindle assembly function during these different stages and transitions remains only partially understood.

XMAP215 and the Transforming and Acidic Coiled-Coil (TACC) are widely conserved proteins, with family members that can bind each other and often function as a complex to regulate microtubule stability in a variety of cellular contexts, including in *C. elegans* oocyte meiotic cell division [15–22]. XMAP215 orthologs have a C-terminal acidic domain that binds microtubules and multiple TOG (Tumor Over-expressed Gene) domains that bind to and increase the local concentration of tubulin heterodimers, modulating their incorporation or

removal at microtubule plus ends [23,24]. Studies in *Xenopus* and in *Drosophila* have identified a C-terminal TACC domain that mediates binding to XMAP215 proteins and promotes the localization of both XMAP215 and TACC proteins to centrosomes [25–27]. During the first mitotic division of the early *C. elegans* embryo, XMAP215/ZYG-9 and TACC/TAC-1 co-localize to centrosomes and spindle microtubules, mutually depend on each other for protein stability, and are both required for spindle microtubule stability and proper mitotic spindle positioning [17,28,29].

While the mitotic functions of ZYG-9 and TAC-1 have been investigated more extensively, their roles during *C. elegans* oocyte meiotic cell division are not as well understood. Early studies showed that ZYG-9 is concentrated at oocyte spindle poles and diffusely associated with spindle microtubules [30], and RNA interference (RNAi) knockdown of ZYG-9 resulted in disorganized spindles and frequent chromosome separation errors [9,30]. More recently, live imaging studies have shown that ZYG-9 and TAC-1 exhibit indistinguishable spindle localization patterns during oocyte meiotic cell division, and RNAi depletion of either protein resulted in identical phenotypes [21]. The earliest detected defects were observed at the microtubule cage stage, when microtubule bundles were not restricted to the periphery but sometimes passed through the internal chromosome-occupied space. Subsequently the coalescence of pole foci was defective, with foci sometimes splitting, and mutant oocytes sometimes forming tripolar spindles that separated chromosomes into three sets.

Notably, in contrast to their well-documented role in promoting microtubule stability during early embryonic mitosis, ZYG-9 and TAC-1 negatively regulate microtubule stability during oocyte meiotic cell division. RNAi knockdown of either results in significantly elevated microtubule levels in oocytes, both in association with the egg chromosomes and throughout the oocyte cortex [21]. Genetic studies in budding yeast [31,32] and in *Drosophila* S2 cells [33], and biochemical studies in *Xenopous* egg extracts [34], have shown that XMAP215 orthologs destabilize microtubules in some contexts. However, most studies have focused on their roles in promoting microtubule stability [22,24,35]. *C. elegans* oocyte meiotic cell division thus provides an appealing model for investigating the negative regulation of microtubule stability by XMAP215 and TACC family members.

The numerous defects observed upon loss of ZYG-9 or TAC-1 suggest that this complex may have multiple separate requirements during oocyte meiotic cell division. However, the use of non-conditional mutations or knockdown methods to assess gene requirements precludes determining whether each defect represents a separate requirement, or if instead the later defects are indirect consequences of earlier ones. The use of CRISPR/Cas9 genome editing to degron-tag genes for auxin-inducible degradation now makes it possible to engineer conditional loss of function alleles throughout the genome [36]. Indeed, a recent study has shown that degron-tagged dynein can be rapidly degraded and knocked down in function within only a few minutes of auxin treatment beginning at metaphase during *C. elegans* oocyte meiosis I, documenting a late role for dynein in the maintenance of spindle structure [37].

Temperature-sensitive (TS) alleles, typically missense mutations that produce functional proteins at lower permissive temperatures but are inactive at higher restrictive temperatures, provide another powerful approach for the temporal dissection of gene requirements. Many *C. elegans* TS alleles can be classified as either slow- or fast-acting [38]. With slow-acting alleles, fertile worms must be cultured for hours at the restrictive temperature to observe mutant phenotypes. Such slow-acting alleles are likely due to mutations that cause irreversible protein folding outcomes and require the production of newly synthesized proteins and oocytes for the consequences of a temperature shift to be observed. By contrast, fast-acting, heat-sensitive TS mutant proteins rapidly inactivate within seconds or minutes of a temperature upshift, even if the proteins were translated at a lower permissive temperature, likely because the

missense mutations make the proteins less thermally stable and more prone to unfolding or inactivation at higher restrictive temperatures. Furthermore, some fast-acting TS alleles also are reversible, with protein function rapidly restored upon shifting from restrictive to permissive temperatures, adding to their usefulness. For example, temperature upshift and downshift experiments have documented earlier and later roles, respectively, for the Aurora B kinase AIR-2 and the centralspindlin component ZEN-4 during mitotic cytokinesis in one-cell stage *C. elegans* embryos [39]. In *C. elegans* oocytes, temperature upshift experiments with TS alleles have been used to document a role for MEI-1/katanin in the maintenance of meiotic chromosome congression [40], and for the kinesin 12 family member KLP-18 in the maintenance of acentrosomal spindle bipolarity [7]. To better define the requirements for ZYG-9 and TAC-1 during oocyte meiotic cell division, we have taken advantage of fast-acting TS alleles, with live imaging and fluorescent protein fusions, to identify multiple and separate meiotic cell division requirements for this widely conserved regulator of microtubule dynamics.

## Results

### Oocyte meiosis I spindle assembly defects in *XMAP215/zyg-9* and *TACC/tac-1* TS mutant oocytes

To temporally dissect requirements for ZYG-9 and TAC-1 during *C. elegans* oocyte meiotic spindle assembly, we have employed two previously isolated recessive and temperature-sensitive alleles, *zyg-9(or634ts)* and *tac-1(or455ts)* [29], and one newly isolated allele, *zyg-9 (or1984ts)* (see Materials and Methods). Hereafter, we collectively refer to the mutant oocytes as *zyg-9* and *tac-1* mutants, or as TS mutants. These alleles all showed less than 1% embryonic lethality when homozygous mutant adults were cultured at the permissive temperature of 15˚C and greater than 99% embryonic lethality when cultured at the restrictive temperature of 26˚C [29] (Table 1), and therefore are likely to be fast-acting [38].

The *zyg-9(or634ts)* and *tac-1(or455ts)* alleles were previously reported to genetically map to the center and right arm of chromosome II, respectively, where the corresponding loci are located, and *zyg-9(or634ts)* failed to complement the previously isolated allele *zyg-9(b244)*

**Table 1. Genetic Analysis of *zyg-9* and *tac-1* TS and Null Alleles.**

| Allele | Homozygote embryonic viability (%) at | | | Heterozygote embryonic viability (%) at | | |
|---|---|---|---|---|---|---|
| | **15˚C** | **20˚C** | **26˚C** | **15˚C** | **20˚C** | **26˚C** |
| Wild type (N2) | 99.6 (n = 987) | 99.4 (n = 1249) | 99.1 (n = 581) | - | - | - |
| **TS allele** | | | | | | |
| tac-1(or455) | 99.8 (n = 768) | 100(n = 622) | 0 (n = 364) | - | - | 97.8 (n = 639) |
| zyg-9(or634) | 97.3 (n = 635) | 91.3(n = 351) | 0 (n = 202) | - | - | 97.1 (n = 763) |
| zyg-9(or1984) | 99.8 (n = 710) | 99.5(n = 615) | 0 (n = 350) | - | - | 99.2 (n = 516) |
| **Deletion allele** | | | | | | |
| tac-1(ok3305) | 0.01 (n = 810) | - | 0 (n = 472) | 99.3 (n = 1187) | 99.2 (n = 2006) | 62.7 (n = 627) |
| zyg-9(or1985) | 0 (n = 1408) | - | 0 (n = 317) | 99.9 (n = 932) | 99.5 (n = 1809) | 67.7 (n = 874) |
| **Complementation test** | | | | | | |
| tac-1(or455)/(ok3305) | - | - | - | 98.9 (n = 529) | - | 0 (n = 286) |
| zyg-9(or634)/(or1985) | - | - | - | 98.3 (n = 422) | - | 0 (n = 540) |
| zyg-9(or1984)/(or1985) | - | - | - | 98.3 (n = 643) | - | 0 (n = 436) |

Embryonic viability in broods produced by hermaphrodites of indicated genotypes cultured at indicated temperatures. The embryonic lethality observed at 26˚C but not at lower temperatures in broods from *ok3305/+* and *or1985/+* hermaphrodites appears to represent maternal-effect haploinsufficiency (see Materials and Methods).

[29]. To further verify that the identified missense mutations in *zyg-9* and *tac-1* are responsible for the phenotypes we have observed at the restrictive temperature, we obtained the previously isolated and likely null allele *tac-1(ok3305)*, and we used CRISPR/Cas9 to generate a *zyg-9* null allele, *or1985*, that removes nearly the entire *zyg-9* open reading frame (S1 Fig; see Materials and Methods). Brood analyses indicated that complete loss of *zyg-9* or *tac-1* function results in a recessive, maternal-effect, embryonic-lethal phenotype, with some haploinsufficiency at 26˚C (Table 1). In genetic crosses the *zyg-9* and *tac-1* TS alleles failed to complement their respective deletion alleles at the restrictive but not at the permissive temperature (Table 1; see Materials and Methods).

To observe oocyte meiotic spindle assembly dynamics in control and TS mutant oocytes, we have used spinning disk confocal microscopy, coupled with a microfluidics temperature-control system, to image live oocytes *in utero* within immobilized whole-mount worms from transgenic strains that express both a GFP-tagged ß-tubulin (GFP::TBB-2) to mark microtubules, and an mCherry-tagged histone (mCherry::H2B) to mark chromosomes (see Materials and Methods). To confirm the presence of the TS mutations, we isolated genomic DNA from each TS mutant strain used for imaging and confirmed the presence of the intended mutant allele (see Materials and Methods). All live imaging was done using oocytes within whole mount worms, which limits resolution but was essential for accurately timing when temperature shifts were done relative to nuclear envelope breakdown (NEBD), the earliest step in oocyte meiotic spindle assembly (S2 Fig and S1 Data). First, we compared oocyte meiotic spindle assembly in control and mutant oocytes maintained at 15˚C throughout meiosis I (see Materials and Methods), to determine if any non-essential defects might occur even when mutant oocytes were kept at the permissive temperature. When maintained at 15˚C, *zyg-9* and *tac-1* mutant oocytes routinely formed barrel-shaped bipolar spindles that aligned chromosomes at the spindle midpoint, with spindle assembly and chromosome separation dynamics indistinguishable from those observed in control oocytes, except for lagging chromosomes during anaphase that we saw in 2 of 20 *zyg-9(or1984ts)* and 1 of 13 *zyg-9(or634ts)* oocytes maintained at 15˚C (Figs 1A–1D and S3). These results indicate that any further defects observed after temperature upshifts are due to inactivation of the mutant protein.

We next assessed how effectively these TS alleles reduce gene function relative to previous studies that used RNAi to knock down ZYG-9 and TAC-1. After TS mutant oocytes were shifted from the permissive to the restrictive temperature of 26˚C shortly before NEBD and then maintained at 26˚C throughout meiosis I (see Materials and Methods), we compared the defects observed during spindle assembly to the defects previously reported following strong RNAi knockdowns [9,21,30]. Control oocytes maintained at 26˚C always became bipolar with normal spindle assembly dynamics (Fig 1E and S1 Movie, 18/18 oocytes). In contrast, spindle microtubules and chromosomes in TS mutant oocytes maintained at 26˚C throughout meiosis I were highly disorganized, with spindles that frequently failed to become bipolar, often failed to separate chromosomes into two sets, and sometimes separated chromosomes into three sets (Figs 1F–1H and S3 and S2 Movies), consistent with previous RNAi knockdown studies [21,30]. To further verify that the defects we observed in the *zyg-9* TS mutant oocytes at the restrictive temperature were caused by loss of ZYG-9 function, we used CRISPR/Cas9 to degron tag the endogenous *zyg-9* locus (see Materials and Methods). After auxin treatment, we observed oocyte meiotic spindle assembly defects that also resembled those reported after RNAi knockdowns (S4 Fig), and that we have observed in TS mutant oocytes at the restrictive temperature (Figs 1 and S3 and S2 Movies). We also examined the expression levels of endogenous GFP fusions to ZYG-9 and TAC-1 after using the same feeding RNAi knockdown conditions we used previously to identify requirements for these two genes [21], and we observed a substantial reduction in GFP signal (S4 Fig), further confirming that the phenotypes observed

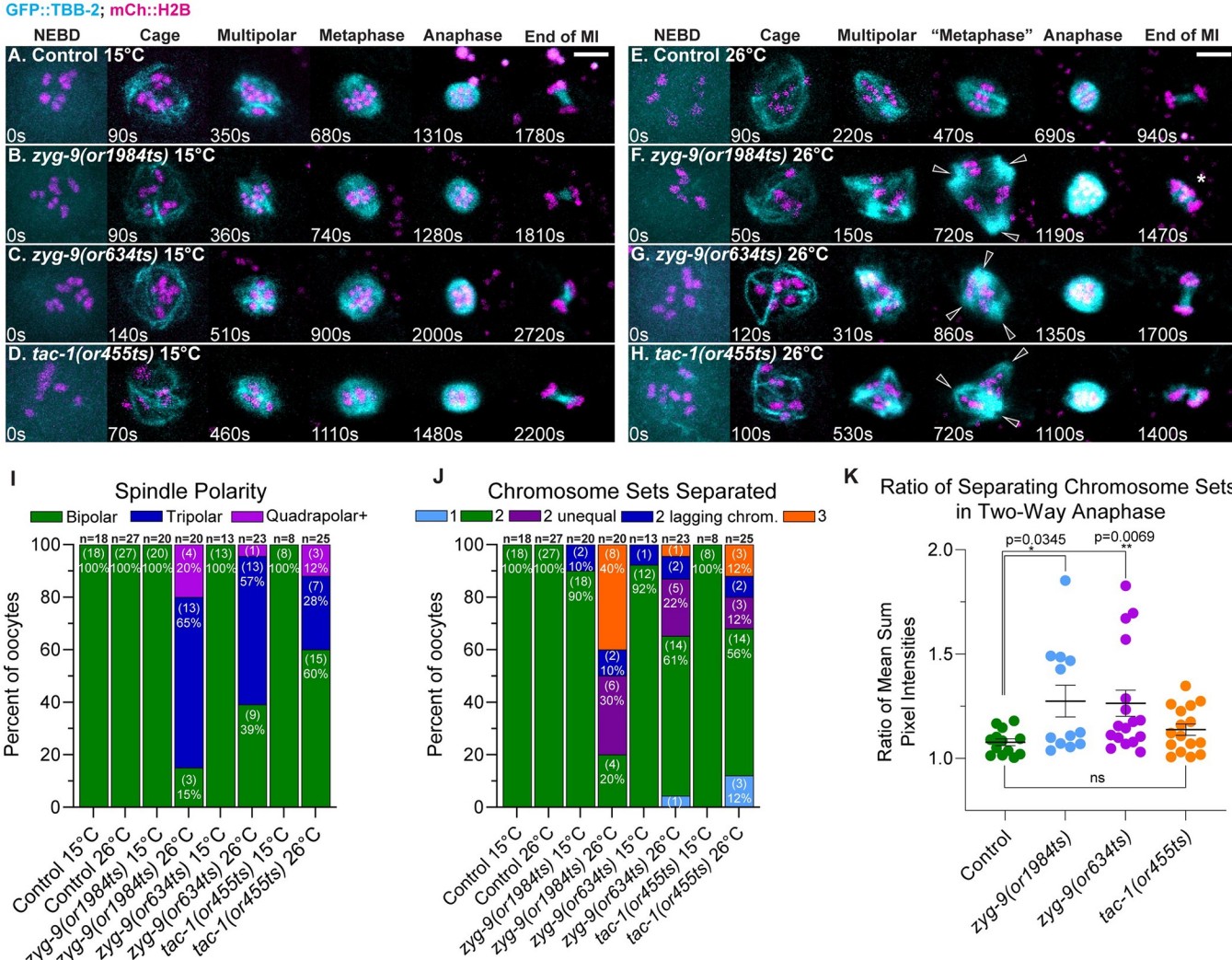

**Fig 1. Oocyte meiosis I spindle assembly dynamics in *zyg-9* and *tac-1* TS mutants at the permissive and restrictive temperatures.** (A-H) Time-lapse maximum intensity projection images during meiosis I in live control and TS mutant oocytes expressing GFP::TBB-2 and mCherry::H2B to mark microtubules and chromosomes, at 15°C (A-D) and at 26°C (E-H). In this and in all subsequent meiosis I time-lapse image series, t = 0 is labeled NEBD and is the timepoint immediately preceding the appearance of microtubule bundles forming the cage structure. To account for differences in movie signal quality inherent to *in utero* live cell imaging, the intensity scales for montages in all figures were set individually to give the clearest depiction of the spindle and chromosomes (see Materials and Methods). (I-J) Number of spindle poles present at the onset of spindle shortening per oocyte (I), and number of chromosome sets separated after anaphase per oocyte (J); number of oocytes examined indicated above each bar, number scored with each phenotype shown in parentheses inside bars with corresponding percent below. (K) Ratio of fluorescence intensity of two separated chromosome sets at 26°C (see 1J) at the end of anaphase B. For all figures, distributions of scatter plot values were compared using the Mann–Whitney U-test to calculate P-values (S2 Data). Error bars and values are mean ± SEM. *, P <0.05; **, P < 0.01. Scale bars = 5 μm. Oocytes with extra poles indicated with hollow arrowheads. See the Materials and Methods for a description of the spindle assembly stages and frame selection in this and all other figures. Oocyte with chromosomes separated into three sets indicated with an asterik.

are due to loss of ZYG-9 function. While these different knockdown methods likely do not entirely eliminate *zyg-9* and *tac-1* function, they all clearly document similar requirements for the encoded proteins during oocyte meiotic spindle assembly.

To more quantitatively compare assembly dynamics and spindle polarity in control and mutant oocytes, we used as a reference point the beginning of spindle shortening, which in wild-type oocytes occurs upon the transition from metaphase to anaphase [9,10]. Even in highly abnormal *zyg-9* or *tac-1* mutant spindles that formed at 26°C, spindle shortening was

easily identifiable (Figs 1F–1H and S3 and S2 Movie), as reported previously after RNAi knockdown [9]. When maintained at 26˚C, spindle bipolarity was established by the beginning of spindle shortening in only 3 of 20 *zyg-9(or1984ts)*, 9 of 23 *zyg-9(or634ts)*, and 15 of 25 *tac-1 (or455ts)* oocytes, whereas in mutant oocytes kept at 15˚C, all spindles were bipolar by the beginning of spindle shortening (Fig 1I).

We also observed severe chromosome separation errors in *zyg-9* and *tac-1* mutant oocytes maintained at the restrictive temperature throughout meiosis I. By late in meiosis I, we often observed separation outcomes that resulted in one or three, instead of two, chromosome sets, as well as lagging chromosomes bridging the sets (Fig 1J). While separation into two sets occurred in roughly half of the TS mutant oocytes, the chromosomal distribution between the two sets was often unequal (Fig 1K), whereas we consistently observed two equally sized chromosome sets in TS mutant oocytes maintained at 15˚C and in control oocytes (Fig 1J and 1K). The penetrance of the severe separation defects—into one or three sets—roughly matched the penetrance of the failures to establish bipolar spindles, suggesting that the defects in spindle assembly may account for subsequent chromosome separation errors. However, while roughly half of mutant oocytes separated chromosomes into three sets after ZYG-9 RNAi knockdown [21], we observed fewer examples (~1/3 overall) of three-way separation in TS mutant oocytes at the restrictive temperature (Fig 1J). We conclude that *zyg-9* and *tac-1* mutant oocytes (i) assemble spindles with roughly normal dynamics at the permissive temperature, and (ii) exhibit spindle assembly and polarity defects at the restrictive temperature very similar to those previously observed after RNAi knockdown, albeit with somewhat lower penetrance.

## TS *zyg-9* and *tac-1* alleles are fast-acting

To determine if these TS *zyg-9* and *tac-1* alleles are fast-acting, we upshifted mutant oocytes from the permissive to the restrictive temperature at the multipolar stage, between roughly 2.7 and 6.8 minutes after cage onset, preceding the establishment of spindle bipolarity (Materials and Methods, S2 Fig). We then observed the subsequent spindle assembly and chromosome separation dynamics. Prior to the upshift, microtubule networks in *zyg-9* and *tac-1* oocytes were coalescing into a bipolar structure as in control oocytes (Figs 2B–2D and S5). Immediately after the upshifts, coalescence was disrupted, and the penetrance of the subsequent spindle assembly and chromosome separation defects were nearly identical to those observed in TS mutant oocytes maintained at 26˚C throughout meiosis I (Figs 1I, 1J, 2I and 2J and control oocyte upshift S3 Movie and *zyg-9(or1984ts)* upshift S4 Movie and *zyg-9(or634ts)* upshift S5 Movie and *tac-1(or455ts)* upshift S6 Movie). In contrast, temperature upshifts at the multipolar stage in control oocytes had no effect on pole coalescence, spindle bipolarity and chromosome separation (Figs 2A and S5; 7/7 oocytes). We conclude that *zyg-9(or1984ts)*, *zyg-9(or634ts)*, and *tac-1(or455ts)* are fast-acting during oocyte meiotic cell division.

## Spindle polarity and chromosome separation defects in *zyg-9* and *tac-1* mutants do not depend on prior cage structure defects

Having established that these TS alleles are fast-acting, we then asked whether the later spindle bipolarity and chromosome separation defects in *zyg-9* and *tac-1* mutant oocytes depend on the abnormal microtubule cage structure that forms shortly after NEBD, the earliest defect described thus far in these mutants (see S7 Movie for control oocyte cage structure). In some *zyg-9* and *tac-1* mutant oocytes, the microtubule bundles that form the cage are not restricted to the periphery as in control oocytes, but also pass through the chromosome occupied space inside the cage [21]. Because the cage structure might promote the coalescence of early pole foci by making the process two dimensional, rather than three-dimensional throughout the

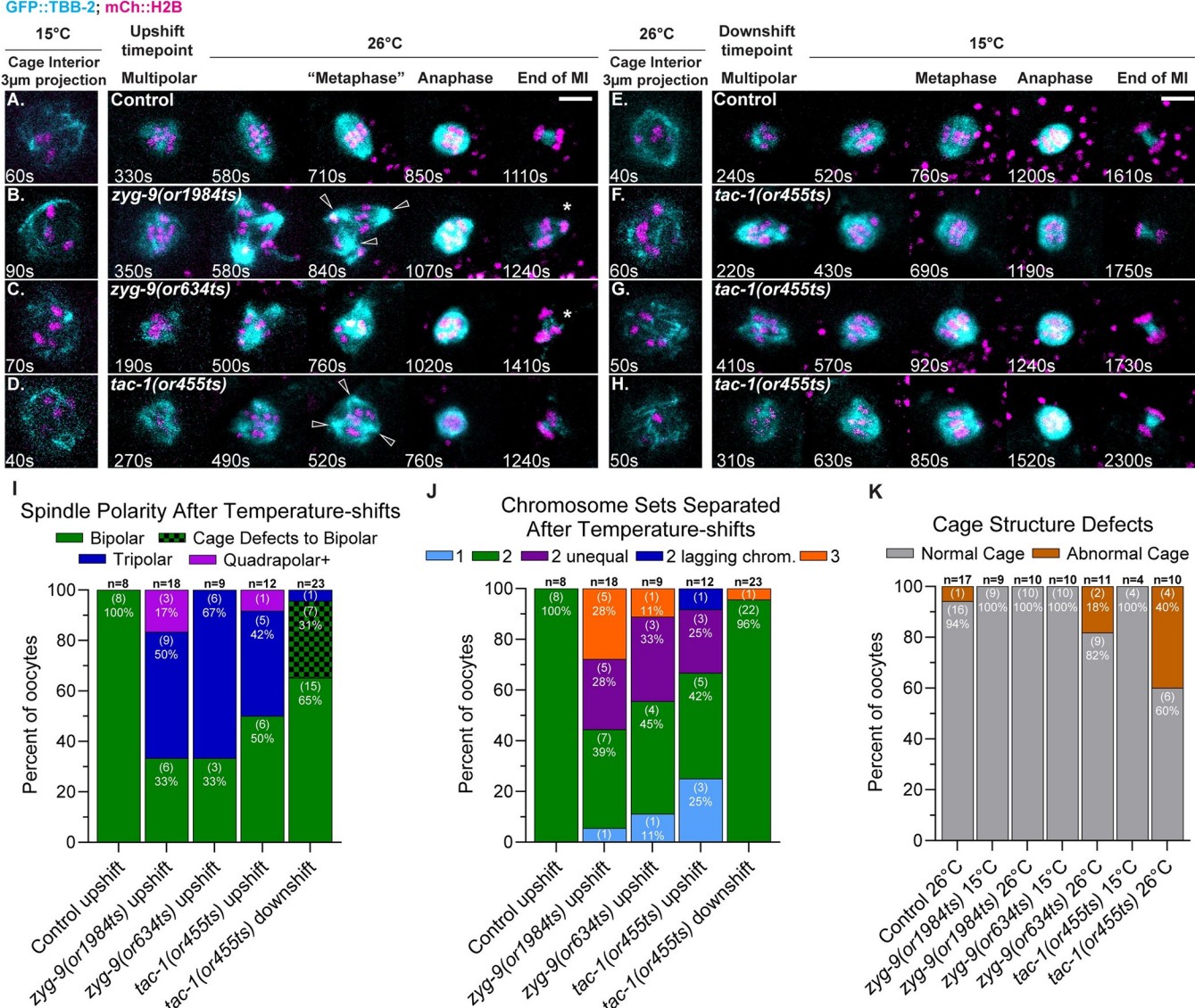

**Fig 2. Pole coalescence and polarity defects in *zyg-9* and *tac-1* mutant oocytes do not depend on prior cage structure defects.** (A-H) Time-lapse maximum intensity projection images of live control and TS mutant oocytes upshifted from 15°C to 26°C (A-D) or downshifted from 26°C to 15°C (E-H) during the multipolar stage, in oocytes expressing GFP::TBB-2 and mCherry::H2B. Oocytes depicted in F-H have cage structure defects (for 3-D images of TS mutant oocytes with cage stage defects, see S9, S10 and S11 Movies). Note that the cage stage images are maximum intensity projections of 3 focal planes (3 μm) that encompass the dis-assembling nucleus, with limited resolution due to in utero imaging. All other images are maximum intensity projections of all 20 focal planes. For images of cage stage control and mutant oocytes that project all 20 focal planes, see S5 Fig. (I-K) Number of spindle poles present at the onset of spindle shortening per oocyte (I); number of chromosome sets separated during anaphase per oocyte (J); number of oocytes with abnormal cage structures (K); number of oocytes examined indicated above each bar, number scored with each phenotype shown in parentheses inside bars with corresponding percent below. Oocytes with extra spindle poles indicated with hollow arrowheads; oocytes with chromosomes separated into three sets indicated with asterisks. Bar graphs in (I) and (J) include oocytes expressing GFP::TBB-2 (this figure) and oocytes expressing GFP::ASPM-1 (Fig 4). Scale bars = 5 μm.

volume occupied by chromosomes, a causal relationship between this cage structure defect and the subsequent pole coalescence defect seems plausible [21]. However, when we examined *zyg-9* and *tac-1* mutant oocytes that failed to establish a bipolar spindle after multipolar stage temperature upshifts, we found that the cage structures had formed properly prior to the upshifts, with microtubule bundles restricted to the periphery (Figs 2B–2D and S5 and S8 Movies for an example of a mutant oocyte with normal cage structure but defective spindle

polarity). These results suggest that the pole coalescence defects in *zyg-9* and *tac-1* mutant oocytes do not depend on an earlier detectably abnormal cage structure.

We next performed complimentary temperature-downshift experiments to ask if bipolar spindle assembly can be rescued in mutant oocytes with earlier cage defects. For these downshift experiments, we used *tac-1(or455ts)*, as it resulted in the most highly penetrant cage structure defect when TS mutants were kept at 26°C throughout meiosis I (Figs 2K and S5G). In *tac-1(or455ts)* oocytes kept at 26°C until the multipolar stage, and then downshifted to 15°C, between roughly 2.3 and 6.2 minutes after NEBD (S2 Fig), the spindle microtubules coalesced after the downshifts into bipolar spindles that separated chromosomes into two equal sets in 22 of 23 mutant oocytes, with the one exception being a mutant oocyte that separated chromosomes into three sets (Fig 2F–2I). Importantly, bipolar spindles assembled and separated chromosomes into two equal sets in all seven of the mutant oocytes in which we observed early cage structure defects, indicating that cage structure defects are not sufficient to cause later coalescence defects (Fig 2J and S9–S11 Movies for examples of TS mutant oocytes with abnormal cage structures that nevertheless formed bipolar spindles and separated chromosomes into sets). Based on these temperature upshift and downshift experiments, we conclude that the pole coalescence defects observed during prometaphase in *zyg-9* and *tac-1* mutants occur independently of the earlier cage structure defects.

## Spindle polarity and chromosome separation defects in *zyg-9* and *tac-1* mutant oocytes do not depend on elevated microtubule levels

Another mutant phenotype in *zyg-9* and *tac-1* mutant oocytes that could indirectly cause defects in spindle bipolarity and chromosome separation is the prominent accumulation of abnormally high levels of microtubules, both in association with oocyte chromosomes and also throughout the oocyte cortex during meiosis I spindle assembly, although the levels vary substantially from oocyte to oocyte, and over time in any one oocyte [21]. To investigate whether increased spindle microtubule levels might be responsible for the failures to establish spindle bipolarity and properly separate chromosomes, we quantified spindle-associated microtubule levels in TS mutants oocytes maintained at 26°C throughout meiosis I (Materials and Methods). Surprisingly, spindle microtubules levels were significantly increased only in *tac-1 (or455ts)* oocytes but not in *zyg-9(or1984ts)* or *zyg-9(or634ts)* oocytes, relative to control oocytes maintained at 26°C (Fig 3A and 3B). Because spindle assembly and chromosome separation were often defective in all three TS mutants when maintained at 26°C, and because the penetrance of the spindle defects were highest in *zyg-9(or1984ts)* and lower in *tac-1(or455ts)* oocytes, a simple elevation in overall spindle microtubule levels alone cannot account for the spindle bipolarity and chromosome separation defects observed after reducing ZYG-9 or TAC-1 function (see Discussion).

## ZYG-9 and TAC-1 prevent the splitting of early pole foci and may limit their growth during bipolar spindle formation

We next used temperature shifts to ask how ZYG-9 and TAC-1 influence the prometaphase coalescence of early pole foci. To assess pole coalescence, we used live imaging of control and mutant oocytes from transgenic strains that express an endogenous GFP fusion to the spindle pole marker ASPM-1 (GFP::ASPM-1), and the mCherry::H2B fusion to mark chromosomes (Figs 4 and S6 and S7). Soon after NEBD in control and mutant oocytes maintained at 15°C throughout meiosis I, diffuse clouds with small GFP::ASPM-1 foci formed a network around the chromosomes, coalescing over time into fewer and larger foci and ultimately forming bipolar spindles with chromosomes aligned midway between the poles (Figs 4A–4C and S6). In

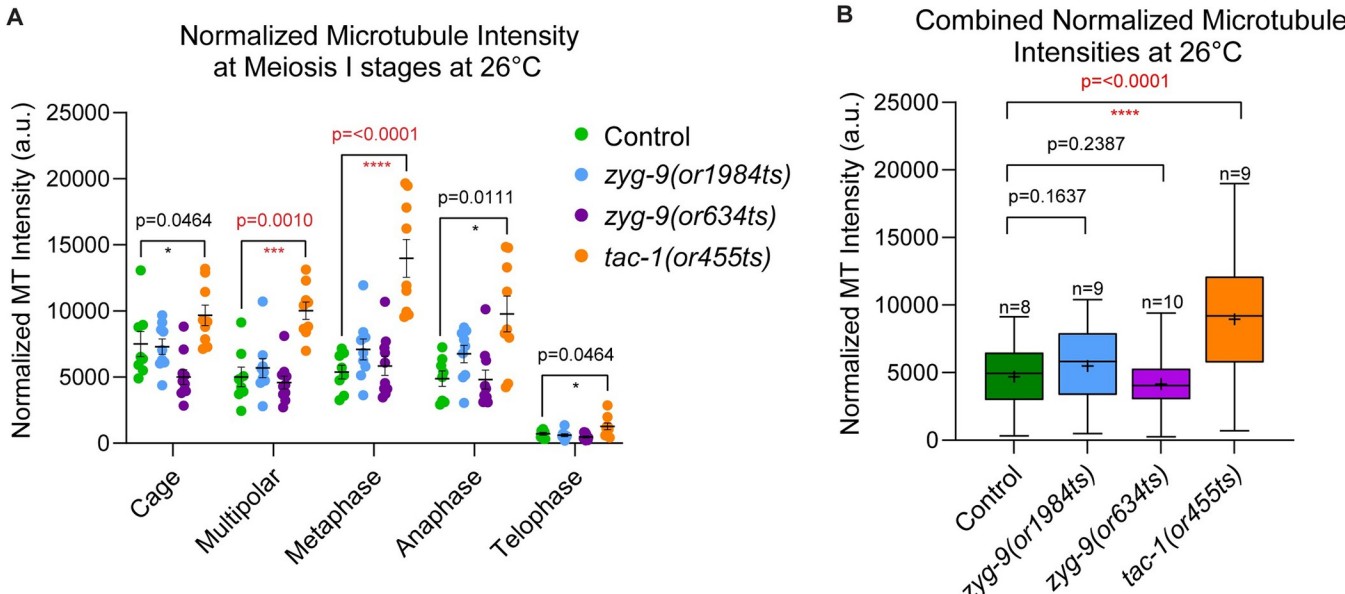

**Fig 3. Microtubule levels are elevated throughout meiosis I in *tac-1(or455ts)* oocytes.** (A-B) Normalized microtubule pixel intensity at 26˚C in arbitrary units (see Materials and Methods) for (A) oocyte spindles at each stage of meiosis I, error bars and values are mean ± SEM; and (B) all meiotic stages combined for control and TS mutant oocytes. The boxplots display the datasets, with the median (line) and mean (+) intensity values for control and mutant oocytes; bars are 75% and whiskers are 95% of the observed normalized microtubule intensity values (S3 Data). *, P <0.05; **, P < 0.01; ***, P < 0.001; ****, P <0.0001.

contrast, in mutant but not control oocytes kept at 26˚C throughout meiosis I, the early pole foci were more dynamic, often splitting apart instead of coalescing, and in some cases the mutant oocytes ultimately assembled spindles with three poles (Figs 4D–4F and S6), consistent with the defects previously reported after RNAi knockdowns [21].

To explore how ZYG-9 and TAC-1 influence pole coalescence, *zyg-9(or1984ts)* and *tac-1 (or455ts)* oocytes expressing the GFP::ASPM-1 pole marker were maintained at 15˚C until the multipolar stage, and then upshifted to the restrictive temperature of 26˚C, as described earlier (S2 Fig). Preceding temperature upshifts in mutant oocytes, the early stages of pole foci coalescence appeared normal, with a diffuse network of small GFP::ASPM-1 foci surrounding the chromosomes. However, upon upshift the early pole foci appeared more dynamic in two ways. First, some pole foci split apart in all TS mutant oocytes after the upshifts, while no foci were observed to split apart after control upshifts (Fig 4H and 4I and Table 2 and S12 Movie for a *tac-1(or455ts)* oocyte). Ultimately, mutant spindles frequently failed to become bipolar, in contrast to upshifted control oocytes (Figs 2I and 4G). Second, pole foci grew more rapidly and to greater size after upshifts in *zyg-9(or1984ts)* oocytes, compared to control oocytes (Figs 4H and 4I and S7 and control oocyte in S13 Movie and *zyg-(or1984ts)* oocyte in S14 Movie). Tracking GFP::ASPM-1 foci revealed that while pole foci were similar in *zyg-9(or1984ts)* and control oocytes preceding upshift, after upshift the foci in *zyg-9(or1984ts)* increased in both volume and intensity more rapidly than in control oocytes, indicated by a significant increase in the slopes of their growth rates and by the mutant foci ultimately becoming larger compared to controls (Figs 4M and 4N; see Materials and Methods). While we did not detect these differences in *tac-1(or455ts)* mutant oocytes (S8 Fig and S15 Movie), the rapid growth of small GFP::ASPM-1 foci into more prominent foci after the temperature upshifts in *zyg-9(or1984ts)* mutant oocytes raises the possibility that ZYG-9 and TAC-1 may promote pole coalescence by limiting the growth of pole foci, in addition to preventing their splitting (see Discussion).

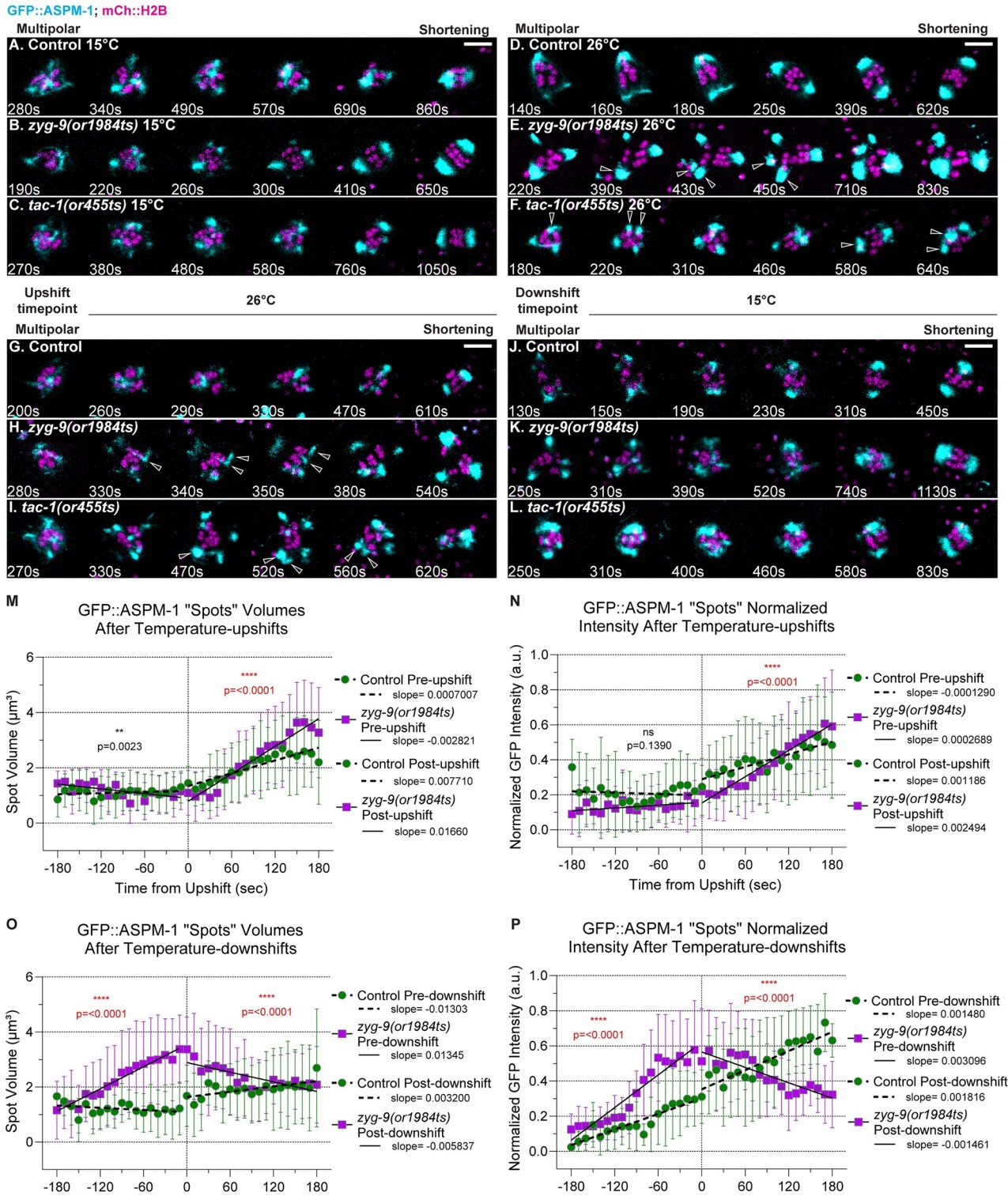

**Fig 4. ZYG-9 and TAC-1 prevent the splitting of early pole foci and limit their growth during meiosis I bipolar spindle assembly.** (A-L) Time-lapse maximum intensity projection images of live control and TS mutant oocytes expressing GFP::ASPM-1 and mCherry::H2B to mark spindle poles and chromosomes, at 15°C (A-C), at 26°C (D-F), and upshifted to 26°C (G-I) or downshifted to 15°C (J-L) during the multipolar stage. Montage frames highlight pole coalescence dynamics during the multipolar stage through to the onset of spindle shortening. Ooctes with extra spindle poles indicated with hollow arrowheads. (M-P) Quantification of control and *zyg-9(or1984ts)* GFP::ASPM-1 foci volume and integrated pixel intensity (see

Materials and Methods) pre- and post-multipolar upshift (M, N) and downshift (O,P). Slopes were compared using a two-tailed t-test to calculate P-values (S4 Data). **, P < 0.01; ****, P <0.0001. Scale bars = 5 μm.

We also performed complimentary multipolar temperature downshift experiments, as described above (S2 Fig), to assess the effects of restoring ZYG-9 and TAC-1 function to a disrupted early network of pole foci. Prior to the downshifts, we observed dynamic and prominent pole foci that grew larger and brighter in *zyg-9(or1984ts)* oocytes compared to controls (Figs 4K and 4L and S7 and control oocyte in S16 Movie and *zyg-9(or1984ts)* oocyte in S17 Movie and *tac-1(or455ts)* in S18 Movie). Upon downshift, prominent pole foci reverted to a more diffuse network of smaller GFP::ASPM-1 foci, with GFP::ASPM-1 focus size and integrated pixel intensity decreasing following the downshift, showing an inverse effect compared to the upshifts, with poles in control oocytes showed a relatively constant growth trajectory (Fig 4O and 4P). Also, nearly all examples of foci splitting in TS mutant oocytes occurred prior to or within 90 seconds of the downshifts (Table 2), indicating that restoring ZYG-9 or TAC-1 function rescues the stability of pole foci during coalescence. Finally, the poles coalesced into bipolar spindles that separated chromosomes into two equal sets in 9 of 10 *zyg-9(or1984ts)* and 6 of 6 *tac-1(or455ts)* downshifted oocytes (Table 2 and S5 Fig). To summarize, these changes in pole dynamics after temperature upshifts and downshifts suggest that ZYG-9 and TAC-1 might influence pole coalescence both by promoting pole stability and by limiting pole growth.

## ZYG-9 and TAC-1 suppress ectopic pole formation and maintain chromosome congression at the metaphase plate

We next asked whether ZYG-9 and TAC-1 are required for pole stability after a bipolar spindle has formed, using temperature upshifts during metaphase in TS mutant oocytes expressing either GFP::ASPM-1 or GFP::TBB-2 and mCherry::H2B. These temperature upshifts were done during an interval ranging from 8.7 to 18 minutes after cage onset and prior to spindle

**Table 2. ZYG-9 and TAC-1 Prevent the Splitting of Early Pole Foci.**

| GFP::ASPM-1 Splitting events post-upshift | |
|---|---|
| Control | 0 of 6 oocytes (0,0,0,0,0,0) |
| *zyg-9(or1984ts)* | 9, in 6 of 9 oocytes (1,2,2,1,0,0,2,0,1) |
| *tac-1(or455ts)* | 10, in 5 of 5 oocytes (1,1,2,2,4) |
| **GFP::ASPM-1 Splitting events up to 90 seconds post-downshift** | |
| Control | 0 of 4 oocytes (0,0,0,0) |
| *zyg-9(or1984ts)* | 10, in 9 of 10 oocytes (1,2,1,1,1,1,1,1,1,0)[a] |
| *tac-1(or455ts)* | 7, in 5 of 6 oocytes (0,1,2,2,1,1)[b] |
| **GFP::ASPM-1 Splitting events >90 seconds post-downshift** | |
| Control | 0 of 4 oocytes (0,0,0,0) |
| *zyg-9(or1984ts)* | 1, in 1 of 10 oocytes (0,0,1,0,0,0,0,0,0,0) |
| *tac-1(or455ts)* | 0 of 6 oocytes (0,0,0,0,0,0) |

*Parentheses show the number of splitting events for each oocyte

[a] 4 of 10 splitting events occurred within 90 seconds after downshift

[b] 3 of 7 splitting events occurred within 90 seconds after downshift

GFP::ASPM-1 splitting events in control and TS mutant oocytes upshifted and downshifted during the multipolar stage.

shortening/anaphase onset (S2 Fig). Following these temperature upshifts, the metaphase poles in TS mutant oocytes remained largely intact, but we nevertheless observed two defects. First, small ectopic poles appeared in the cytoplasm near the spindle in 5 of 17 *zyg-9(or1984ts)*, 8 of 12 *zyg-9(or634ts)*, and 8 of 18 *tac-1(or455ts)* oocytes, while no such ectopic poles were observed in 19 control oocytes (Figs 5A–5D, 6A–6D, S9 and S10 and *zyg-9(or634ts)* oocyte in S19 Movie). The ectopic poles in the mutant oocytes varied in size, were mobile and often fused with one of the two previously established poles before spindle shortening (Fig 5L).

The second defect we observed in mutant oocytes after metaphase upshifts was a failure to maintain the congression of individual bivalents (paired homologs) at the metaphase plate. Individual bivalents moved away from the metaphase plate in 4 of 17 *zyg-9(or1984ts)*, 3 or 12 *zyg-9(or634ts)*, and 4 of 18 *tac-1(or455ts)* metaphase-upshifted oocytes (Figs 5E–5G, 5I–5K, 6B–6H and S9 and S10 and *zyg-9(or634ts)* oocyte in S20 Movie). Importantly, the poorly congressed bivalents were associated with ectopic poles in 9 of the 11 TS mutant oocytes with congression errors (Figs 5M and S10 and *zyg-9(or1984ts)* oocyte in S21 Movie). Furthermore, in upshifted mutant oocytes expressing GFP::TBB-2 and mCherry::H2B and exhibiting congression defects, we observed ectopic microtubule bundles that extended toward poorly congressed bivalents, and spindle poles that appeared partially split (Figs 6G and 6H and S10 and *tac-1(or455ts)* oocyte in S22 Movie). These results are consistent with the hypothesis that either de novo ectopic pole formation, or pole instability, disrupts bipolar spindle structure and hence bivalent alignment. We therefore suggest that ZYG-9 and TAC-1 act during metaphase to suppress ectopic pole formation, and that this pole stability is important for maintaining chromosome congression.

We also asked whether the extensive anaphase chromosome separation defects observed in TS mutant oocytes kept at 26˚C throughout meiosis I are due to earlier pole coalescence and pole stability defects, or alternatively if ZYG-9 and TAC-1 might also have more direct roles in chromosome separation during anaphase. To distinguish between these two possibilities, we analyzed chromosome separation in TS mutant oocytes after metaphase upshifts and found that meiosis I chromosome separation outcomes resulting in just one or in three chromosome sets were nearly eliminated, suggesting that the earlier pole coalescence defects caused these more severe defects in chromosome separation (Fig 5N). However, chromosome separation outcomes with two unequal chromosome sets were still observed in 2 of 17 *zyg-9(or1984ts)*, 1 of 12 *zyg-9(or634ts)*, and 4 of 18 *tac-1(or455ts)* oocytes (Fig 5N and 5O) and defects in the maintenance of chromosome congression accounted for all observed examples of chromosomes separating into two unequal sets (S19 and S20 Movies). These results suggest that the earlier defect in pole coalescence and the later defect in pole stability together account for all defective chromosome separation outcomes. While we did not detect any further and possibly more direct roles for this protein complex during chromosome separation after metaphase upshifts, the modestly lower penetrance of the defects observed in TS mutant oocytes maintained at the restrictive temperature throughout meiosis I, compared to those observed after RNAi knockdown [21], indicates that these TS alleles do not fully reduce gene function at the restrictive temperature. It also is possible that the later upshifts did not allow for enough time to reduce ZYG-9 and TAC-1 function, compared to the earlier upshifts, preventing us from detecting any later requirements.

## ZYG-9 and TAC-1 are required for anaphase spindle rotation and polar body extrusion

Previous studies of early embryonic mitosis have shown that loss of ZYG-9 or TAC-1 results in both short astral microtubules and abnormal mitotic spindle orientation due to the loss of

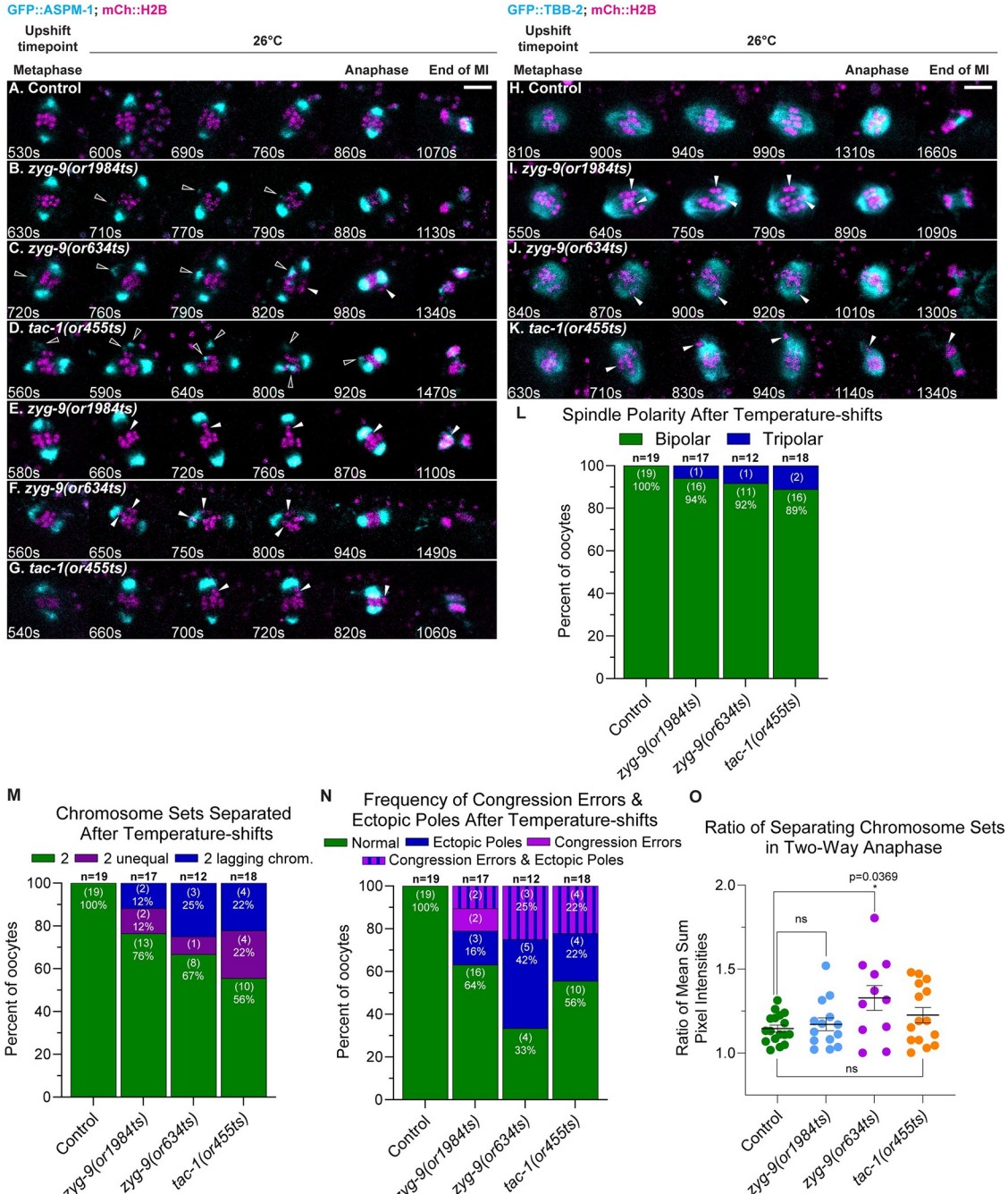

**Fig 5. ZYG-9 and TAC-1 suppress ectopic spindle pole formation and maintain chromosome congression during meiosis I metaphase.** (A-K) Time-lapse maximum intensity projection images of live control and TS mutant oocytes upshifted at metaphase and expressing either GFP::ASPM-1 and mCherry::H2B (A-G) or GFP::TBB-2 and mCherry::H2B (H-K). Montage frames highlight defects following metaphase upshift through to the end of meiosis I. Hollow arrowheads indicate ectopic spindle poles and solid arrowheads indicate chromosome congression errors. (L-N) Number of spindle poles present at the onset of spindle shortening in metaphase upshifted oocytes (L); of chromosome congression errors (M); of ectopic poles in metaphase upshifted oocytes (M); of chromosome sets separated during anaphase (N) for metaphase upshifted oocytes; number of oocytes examined indicated above each bar, number scored with each phenotype shown in parentheses inside bars with corresponding percent below. (O) Ratio of fluorescence intensity of two separated chromosome sets in metaphase upshifted oocytes (see 5N) at the end of anaphase B. *, P <0.05 (S2 Data). Scale bars = 5 μm.

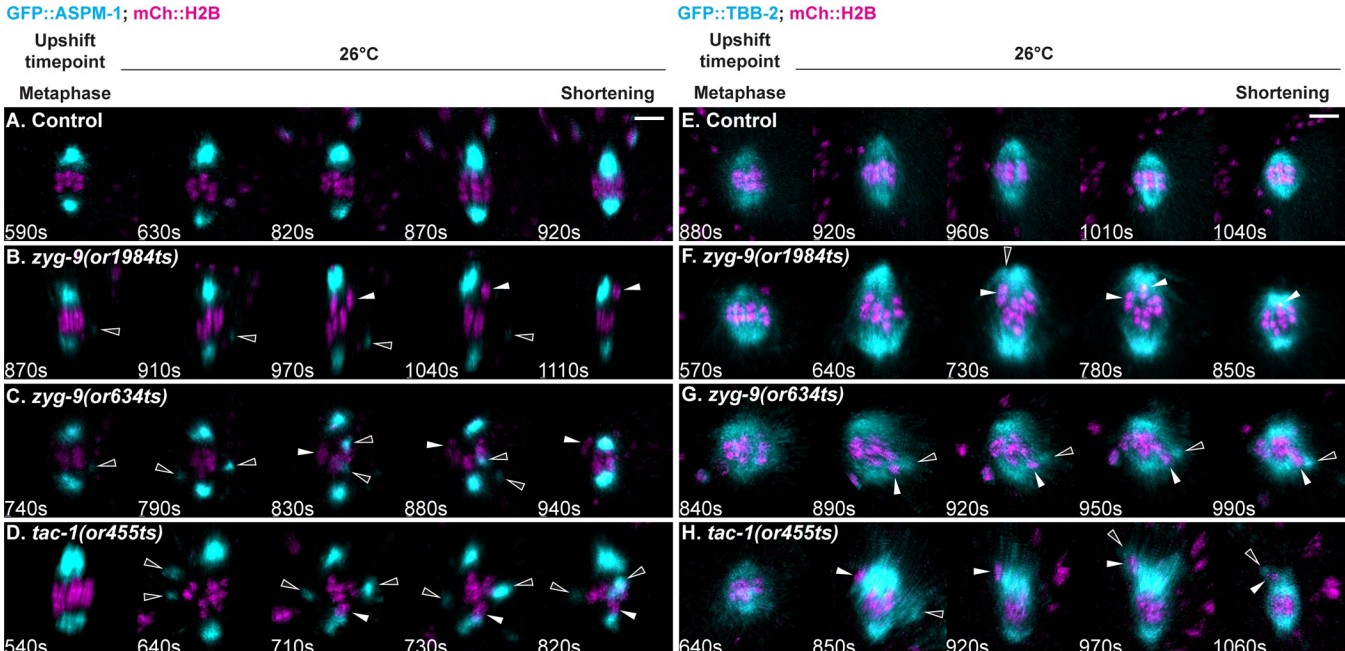

**Fig 6. Ectopic poles are associated with chromosome congression defects in *zyg-9* and *tac-1* TS mutant oocytes.** (A-H) Imaris rotated and snapshot projected time-lapse images (see Materials and Methods) of live control and TS mutant oocytes upshifted at metaphase expressing GFP::ASPM-1 and mCherry::H2B (A-D) or GFP::TBB-2 and mCherry::H2B (E-H). Montage frames highlight defects following metaphase upshift through to spindle shortening. Hollow arrowheads indicate ectopic spindle poles; solid arrowheads indicate chromosome congression errors. Montages of oocytes in A, C, D, F, G, and H are the same oocytes shown in Fig 5 montages A, C, D, I, J, and K, respectively. Scale bars = 5 μm.

uniform astral microtubule contact with the cell cortex [17,29]. To determine if ZYG-9 and TAC-1 also influence meiotic spindle positioning, we scored spindle orientation relative to the overlying cell cortex in *zyg-9* and *tac-1* mutant oocytes after metaphase upshifts to the restrictive temperature. In control upshifted oocytes, the spindle shortened in the pole-to-pole axis and then rotated to become roughly perpendicular to the cortex, such that three or more of the six bivalents contacted the cortex prior to chromosome separation (Fig 7A and 7E; 19/19 oocytes), one measure of proper spindle rotation [9,10,41,42]. In contrast, spindles failed to rotate, or only partially rotated, in 4 of 13 *zyg-9(or1984ts)*, 8 of 12 *zyg-9(or634ts)*, and 8 of 18 *tac-1(or455ts)* oocytes (Figs 7B–7E and S11 and *tac-1(or455ts)* oocyte in S23 Movie). We also scored spindle orientation by measuring the angle of the spindle axis relative to a tangent of the cortex after rotation [42]. While all control oocytes rotated to within an 80–90˚ range, most of the spindles in *zyg-9* and *tac-1* mutant oocytes failed to fully rotate (Fig 7F).

Because spindle rotation is thought to facilitate extrusion of excess oocyte chromosomes into a polar body, we also scored whether chromosomes were successfully extruded. In control upshifted oocytes, chromosomes were always detected within an external polar body at the onset of meiosis II (n = 14). By contrast, polar body extrusion was defective after metaphase upshifts in 5 of 18 *zyg-9(or1984ts)*, 3 of 7 *zyg-9(or634ts)*, and 8 of 14 *tac-1(or455ts)* oocytes, with all chromosomes present within the oocyte cytoplasm at the beginning of meiosis II (Figs 7B–7E and 7G and S11 and *zyg-9(or634ts)* oocyte in S24 Movie). While polar body extrusion was more likely to fail in oocytes with spindle rotation defects, the correlation was only partial (S11I Fig), suggesting that other defects may contribute to the extrusion failures.

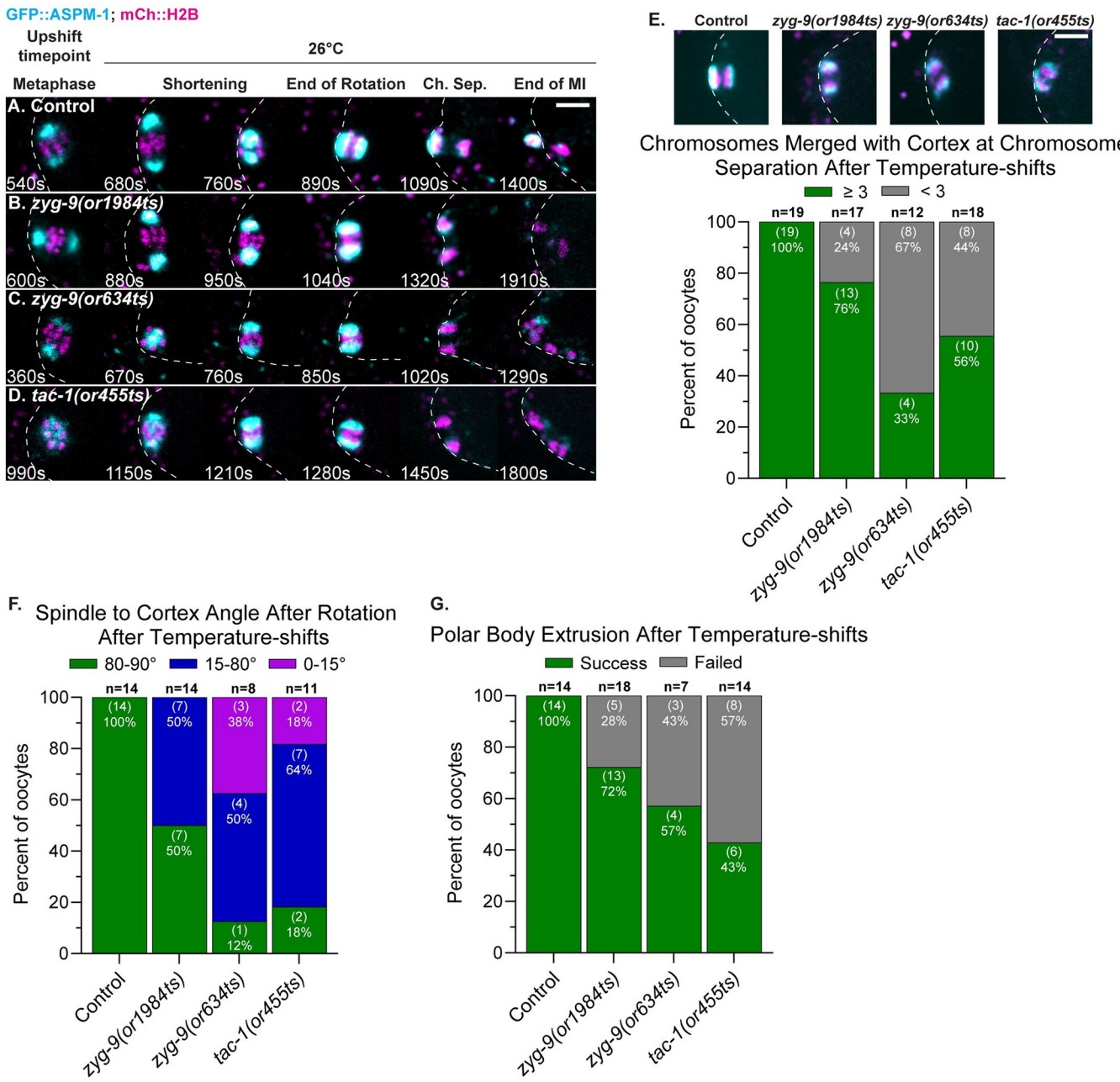

**Fig 7. ZYG-9 and TAC-1 are required for meiosis I anaphase spindle rotation and polar body extrusion.** (A-D) Time-lapse maximum intensity projection images of live control and TS mutant oocytes expressing GFP::ASPM-1 and mCherry::H2B upshifted at meiosis I metaphase; polar body extrusion failed in all three mutant oocytes. Oocytes were rotated so that the cortex was positioned to the left of the spindle; dashed lines depict the oocyte cortex. (E) Early anaphase time-lapse images of live control and TS mutant oocytes after metaphase upshifts. Bar graph quantifies spindle rotation based on the number of chromosomes adjacent to the cortex at the start of chromosome separation in metaphase upshifted oocytes (see Materials and Methods). (F) Spindle angles relative to the cortex at early anaphase in metaphase upshifted oocytes. (G) Number of control and TS mutant metaphase upshifted oocytes that extrude a polar body during meiosis I (see Materials and Methods). Scale bars = 5 μm.

## ZYG-9 and TAC-1 also promote pole coalescence during meiosis II spindle assembly

Many of the proteins required for oocyte meiosis I are present during and thought to be required for meiosis II, although evidence for this is lacking as meiosis I defects can preclude

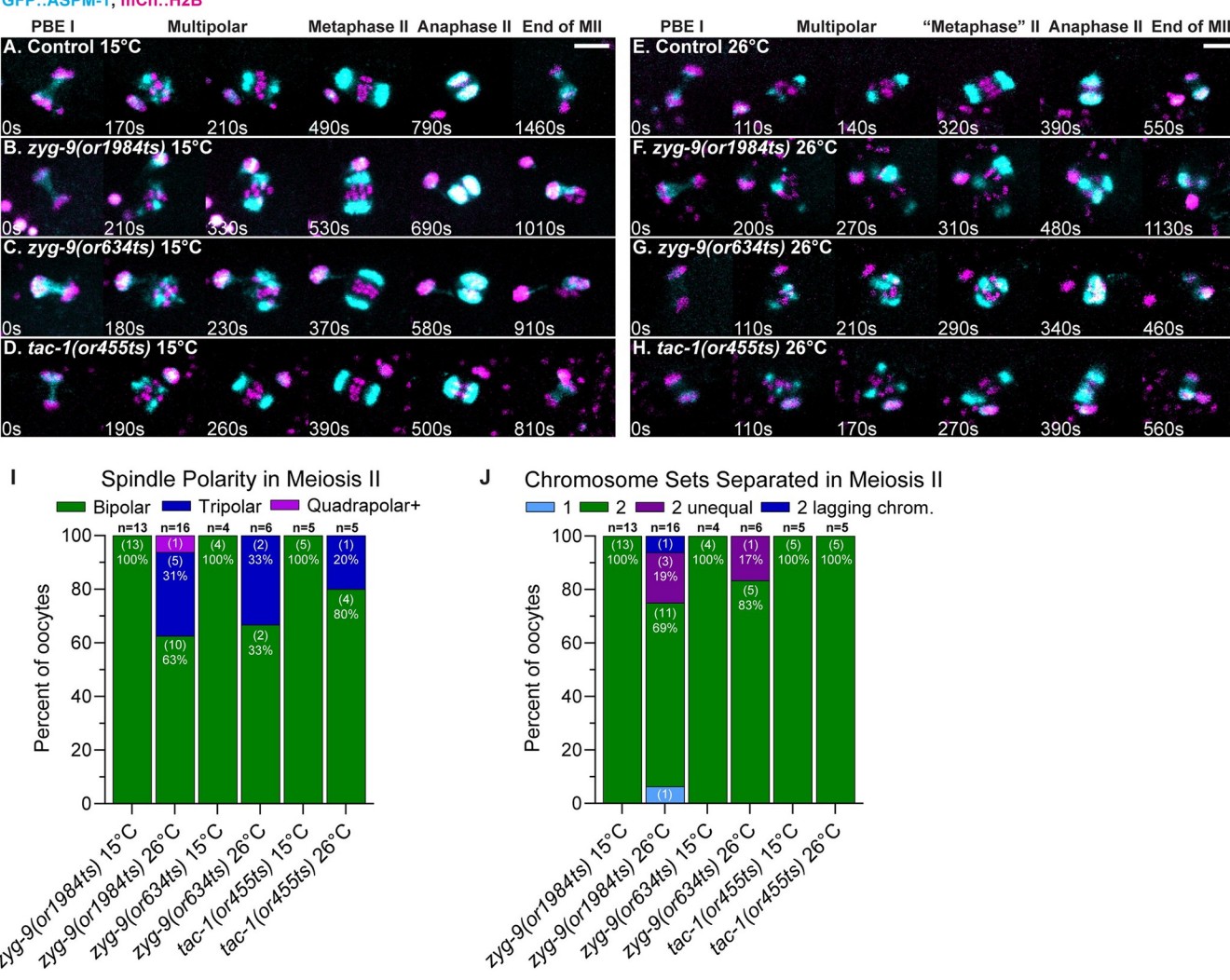

**Fig 8. ZYG-9 and TAC-1 are required for meiosis II spindle assembly.** (A-H) Time-lapse maximum intensity projection images during meiosis II of live control and TS mutant oocytes expressing GFP::ASPM-1 and mCherry::H2B at 15°C (A-D) and at 26°C (E-H). t = 0 is the timepoint when meiosis I chromosome separation ends. (I-J) Number of spindle poles present per oocyte at the onset of spindle shortening (I); of chromosome sets separated during anaphase II (J); number of oocytes examined indicated above each bar, number scored with each phenotype shown in parentheses inside bars with corresponding percent below. Scale bars = 5 μm.

the assessment of later requirements. We therefore used our fast-acting TS alleles to ask whether ZYG-9 and TAC-1 are required for meiosis II spindle assembly. The second meiotic division begins immediately after extrusion of the first polar body, and spindle assembly largely resembles meiosis I except that (i) the spindle is smaller and the duration of meiosis II shorter [43], and (ii) the nuclear envelope has already disassembled, with microtubules appearing amongst the chromosomes instead of forming a peripheral cage structure [7]. To view normal meiosis II pole assembly dynamics, we imaged control and TS mutant oocytes from strains expressing GFP::ASPM-1 and mCherry::H2B and maintained at 15°C. Meiosis II began with small GFP::ASPM-1 pole foci forming around the chromosomes and then coalescing into a bipolar spindle with pairs of sister chromatids aligned between the poles (Figs 8A–8D and S12). We next upshifted control and TS mutant oocytes to the restrictive temperature during meiosis I polar body extrusion, to inactivate ZYG-9 and TAC-1 during meiosis II after

bypassing the meiosis I requirements. Spindle pole foci in *zyg-9* and *tac-1* upshifted oocytes also appeared around the chromosomes but were more dynamic and took longer or failed to assemble into bipolar spindles (Figs 8F–8H and S12). While the spindle bipolarity and chromosome separation defects in all three mutants were reduced in penetrance during meiosis II compared to meiosis I (Figs 1I and 1J and 8I and 8J), these results indicate that ZYG-9 and TAC-1 also influence pole coalescence and spindle bipolarity during meiosis II. As these TS mutations do not appear to reduce gene function as strongly as do RNAi knockdowns, further analysis of the degron-tagged *zyg-9* allele might reveal more penetrant and severe defects during meiosis II.

## Discussion

Gene requirements for the acentrosomal assembly of oocyte meiotic spindles have been investigated to varying extents in different model organisms [2–5]. Experiments in *Drosophila*, *C. elegans* and vertebrate oocytes have shown that XMAP215 orthologs, and their TACC-domain containing binding partners, have important roles in acentrosomal oocyte spindle assembly. Non-conditional RNAi knockdowns in *C. elegans* have shown that XMAP215/ZYG-9 and TACC/TAC-1 are required for multiple aspects of meiosis I spindle assembly: preventing microtubule bundles from crossing through the space occupied by oocyte chromosomes after nuclear envelope breakdown, limiting the overall levels of spindle- and cortex-associated microtubules, and promoting the coalescence of early pole foci into a bipolar structure that separates chromosomes into two equal sets. Our results, using fast-acting TS alleles of *zyg-9* and *tac-1*, with live imaging and fluorescent protein fusions, indicate that the later pole coalescence defects in *zyg-9* and *tac-1* mutants occur independently of the earlier defect in microtubule cage structure, and that ZYG-9 and TAC-1 also are required for pole coalescence during meiosis II. Our results further suggest that ZYG-9 and TAC-1 act during metaphase to promote acentrosomal pole stability, and that this metaphase pole stability contributes to the maintenance of chromosome congression at the metaphase plate. Another recent study also indicates that ZYG-9 and TAC-1 are required for pole stability during oocyte meiosis I, and that ZYG-9 and TAC-1 are codependent each other for their localization to oocyte meiotic spindle poles [44].

### ZYG-9 and TAC-1 have multiple and temporally distinct requirements during oocyte meiotic spindle assembly

Our use of fast-acting TS alleles has identified at least three temporally separable requirements for ZYG-9 and TAC-1 during oocyte meiotic cell division. First, ZYG-9 and TAC-1 limit the appearance of microtubule bundles to the periphery after nuclear envelope breakdown, such that they form a roughly spherical cage that surrounds the oocyte chromosomes early in spindle assembly. Second, our temperature upshift and downshift experiments indicate that the later defects in pole coalescence do not depend on an earlier cage stage defect. Following temperature upshifts during prometaphase pole coalescence, mutant oocytes exhibited polarity defects even in the absence of any pre-existing cage defect. Furthermore, following temperature downshifts at the prometaphase stage, mutant spindles with early cage defects formed bipolar spindles that separated chromosomes normally into two equally sized sets, indicating that a pre-existing cage defect is not sufficient to cause later defects in pole coalescence. We conclude that the defective coalescence of pole foci in TS mutant oocytes represents a distinct and later requirement for this protein complex during the coalescence process itself. This finding further indicates that the mutant spindle polarity defects are not due to coalescence occurring in three dimensions, rather than within the more restricted two-dimensional surface normally defined by the microtubule bundles that form the early cage structure, as we previously speculated [21]. Finally, following temperature upshifts after the completion of meiosis I, we also observed spindle

assembly and pole coalescence defects during meiosis II, a third temporally distinct requirement. We conclude that this protein complex is active throughout meiosis I and II and performs multiple temporally separable and therefore more direct roles.

We also have shown that the pole stability defects in *zyg-9* and *tac-1* mutant oocytes appear to be independent of the overall increase in spindle-associated microtubule levels observed after ZYG-9 or TAC-1 RNAi knockdowns [21]. While we documented similar pole coalescence and chromosome separation defects at the restrictive temperature in all three TS alleles, spindle-associated microtubule levels were reproducibly elevated only in *tac-1(or455ts)* oocytes, which exhibited lower penetrance with respect to spindle polarity defects, indicating that the pole coalescence defects do not depend on an overall increase in microtubule levels. However, these results do not rule out more localized or temporally restricted roles for destabilizing microtubules during pole coalescence and spindle assembly (see below). Finally, the requirements for ZYG-9 and TAC-1 to limit oocyte spindle and cortex microtubule levels during meiosis I further indicates that this protein complex likely functions throughout much of oocyte meiotic cell division to influence microtubule stability and cell division.

## An emerging role for the regulation of microtubule dynamics during polar body extrusion

We have shown that both rotation of the oocyte spindle to orient its axis perpendicular to the overlying cell cortex, and the extrusion of chromosomes into a polar body, often fail in *zyg-9* and *tac-1* mutant oocytes. These defects were observed after metaphase upshifts and thus appear to be independent of the earlier cage structure and prometaphase pole coalescence defects. *C. elegans* oocyte meiotic spindle rotation is thought to require sparse and short astral microtubules, microtubule motors, and shortening of the spindle into a near-spherical shape [41,42,45]. Defects in spindle structure or dynamics that occurred after the metaphase upshifts and prior to rotation and extrusion, including the transient appearance of ectopic pole foci and misshapen spindle poles, might indirectly account for the rotation and extrusion defects.

While spindle rotation defects might indirectly cause failures in polar body extrusion, ZYG-9 and TAC-1 could have more direct roles. Contractile ring assembly and ingression have not been examined in mutants with oocyte spindle rotation defects, and whether rotation is required for proper extrusion is not known [46]. Furthermore, the lack of a clear correlation between failed rotation and failed extrusion after metaphase upshifts in TS mutant oocytes, with polar body extrusion sometimes failing after apparently normal rotation, suggests that ZYG-9 and TAC-1 could have additional and more direct roles in polar body extrusion. Recent studies of the *C. elegans* TOG domain containing protein and CLASP2 family member CLS-2 indicate that proper regulation of microtubule stability is important for contractile ring assembly and ingression during polar body extrusion [46]. However, in contrast to *zyg-9* and *tac-1* mutant oocytes, in which strong loss of function results in abnormally high levels of spindle- and cortex-associated microtubules during oocyte meiotic cell division, loss of CLS-2 results in reduced oocyte spindle and cortical microtubule levels. It will be interesting to further explore the requirements for ZYG-9 and TAC-1 during spindle rotation and polar body extrusion, with later temperature upshifts and downshifts, or acute auxin-induced degradation of degron-tagged ZYG-9, along with live imaging of contractile ring assembly and ingression.

## ZYG-9 and TAC-1 prevent the splitting and may limit the growth of pole foci during prometaphase pole coalescence

Our temperature upshift and downshift experiments indicate that ZYG-9 and TAC-1 act during prometaphase to promote the coalescence of pole foci into a bipolar spindle. Temperature

upshifts during pole coalescence, with live imaging of the GFP::ASPM-1 spindle pole marker, resulted in two informative changes in pole dynamics. First we often saw early GFP::ASPM-1 pole foci split in two after pole coalescence upshifts in TS mutant oocytes, while we did not observe such splitting after control upshifts. We conclude that ZYG-9 and TAC-1 promote the fusion of early pole foci in part by opposing their fission, with fusion over time reducing their number and resulting in bipolarity. We were surprised to also observe a second change in *zyg-9(or1984ts)* oocytes: a rapid increase in both the average size and the integrated pixel intensity of the GFP::ASPM-1 foci after upshifts during pole coalescence, and a corresponding decrease in their size and intensity after pole coalescence downshifts.

It is important to note that we observed these effects on pole growth dynamics only after temperature shifts in *zyg-9(or1984ts)* and not in *tac-1(or455ts)* oocytes, while we observed the splitting of pole foci, failures to establish spindle bipolarity, and failures to properly separate chromosomes after both *zyg-9(or1984ts)* and *tac-1(or455ts)* upshifts. Our failure to detect such changes in ASPM-1 pole foci dynamics in *tac-1(or455ts)* may indicate that such changes in foci dynamics are not required for the spindle and chromosome separation defects that *zyg-9* and *tac-1* TS mutants both share. However, the spindle bipolarity and chromosome separation defects are more highly penetrant in *zyg-9(or1984ts)* than in *tac-1(or455ts)* oocytes, and it is possible that more complete elimination of TAC-1 might reveal similar detectable changes in pole foci dynamics and more penetrant spindle defects. Moreover, it also is possible that the mutation in TAC-1, while not affecting pole foci growth dynamics, may disrupt another function that also is required for spindle assembly and function. Indeed, the higher penetrance of the spindle polarity and chromosome separation defects in *zyg-9(or1984ts)* oocytes, and the higher penetrance of the cage defect in *tac-1(or455ts)* oocytes, suggest that these mutations differently alter complex function. Alternatively, the change in pole foci growth dynamics might reflect properties of ZYG-9 that are not related to the spindle assembly and chromosome separation defects we and others have documented in *zyg-9* mutant oocytes, and it is further possible that tightly linked second site mutations outside of the *zyg-9* locus are responsible for the altered dynamics in pole growth. We speculate that these conditional and partial loss-of-function alleles differently alter complex function, and that the failure to detect the changes in pole foci growth properties associated with the weaker partial loss-of-function *tac-1* mutation does not preclude a role in limiting the growth of pole foci being important for pole coalescence and stability.

## ZYG-9 and TAC-1 promote the maintenance of spindle bipolarity and chromosome congression during metaphase

We observed two additional and correlated phenotypes upon metaphase upshifts in TS mutant oocytes, after bipolar oocyte spindles had already assembled. Ectopic pole foci often appeared near pre-existing bipolar spindles, and individual bivalents often migrated away from the metaphase plate toward one pole. While we did not directly observe any splitting of spindle poles to produce these ectopic foci, we cannot rule out such fission as their source, as opposed to de novo assembly. Notably, these ectopic foci were almost always observed in association with bivalents that moved away from the metaphase plate: in 9 of the 11 oocytes with bivalents that became displaced towards one pole after the metaphase upshifts, we also observed an ectopic focus of GFP::ASPM-1, or an ectopic bundle of spindle microtubules. We speculate that similar such defects escaped detection, due to the limits of our imaging methods, but were also associated with displaced bivalents in the remaining two oocytes. We conclude that this protein complex promotes both the prometaphase coalescence of pole foci and the subsequent stability of spindle poles during metaphase. Another recent study has shown that the minus-

end directed microtubule motor dynein also is required for the maintenance of spindle pole integrity in *C. elegans* oocytes [37]. Whether dynein and ZYG-9/TAC-1 function in the same or different pathways to promote pole stability requires further investigation.

## Microtubule elongation in opposition to acentrosomal spindle pole coalescence

Presumably the prometaphase pole coalescence and metaphase pole stability defects in *zyg-9* and *tac-1* mutant oocytes reflect a requirement for this protein complex in the regulation of microtubule dynamics (see Introduction). Most studies of XMAP215 and TACC orthologs have focused on their roles in promoting microtubule stability as microtubule polymerases [24,35,47,48]. However, ZYG-9 and TAC-1 appear at least in sum to promote microtubule instability during oocyte meiotic cell division [21], with loss of their function resulting in elevated oocyte microtubule levels, both in association with the meiotic spindle and throughout the oocyte cortex. This is in substantial contrast to most studies of XMAP215 and TACC orthologs, and to ZYG-9 and TAC-1 during mitosis in the early *C. elegans* embryo, when loss of their function results in an overall decrease in microtubule stability (see Introduction).

While our results indicate that the pole coalescence and stability defects are independent of an overall increase in microtubule levels during oocyte meiotic cell division (see above), we nevertheless speculate that spatially and temporally restricted limits to microtubule elongation might underlie the defects in pole coalescence and stability observed in mutant oocytes. For example, XMAP215 orthologs are enriched at meiotic and mitotic spindle poles, where they have been proposed to both stabilize microtubule minus ends and counteract the activity of microtubule depolymerases at plus ends [26,27,49]. If ZYG-9 and TAC-1 can also act at pole foci, but to destabilize minus ends or limit microtubule polymerization at plus ends, localized and excessive microtubule growth in mutant oocytes might destabilize coalescing pole foci, with the forces generated by microtubule elongation pushing foci apart. Such spatially and temporally localized perturbations to microtubule dynamics in the TS mutant oocytes might not be reflected in overall microtubule levels. Similarly, excessive microtubule polymerization from spindle poles during metaphase might also promote pole instability and thereby generate ectopic pole foci that mislocalize congressed bivalents. It is perhaps less clear how limiting the rate of pole foci growth might promote coalescence, but excessively rapid foci growth might similarly promote locally excessive microtubule polymerization in mutant oocytes.

## Liquid-liquid phase transitions and acentrosomal spindle pole coalescence

ZYG-9 and TAC-1 orthologs can undergo liquid-liquid phase transitions, and such condensates might mechanistically contribute to acentrosomal pole coalescence and stability in *C. elegans* oocytes. Recent studies in non-human mammalian oocytes have shown that XMAP215/chTOG and TACC/TACC3 exhibit liquid droplet properties during oocyte meiotic spindle assembly [22]. Intriguingly, *C. elegans* XMAP215/ZYG-9, in association with *in vitro* condensates of the coiled-coil centrosome scaffolding protein SPD-5, can concentrate tubulin and promote microtubule nucleation in the absence of any other factors [50]. However, *C. elegans* ZYG-9 and TAC-1 are generally enriched at oocyte meiotic spindle poles and also broadly associated with spindle microtubules [21], but do not appear to exhibit any of the condensate dynamics observed for chTOG and TACC3 in non-human mammalian oocytes; such condensates also are not observed in human oocytes [22].

Liquid droplet properties of spindle proteins during oocyte meiotic spindle assembly have not been documented in *C. elegans*, but pole foci coalescence with the GFP::ASPM-1 marker does resemble condensate fusion. Notably, depletion of TACC3—but not chTOG—from

mouse oocytes disrupts the ability of other spindle proteins to form condensates, with liquid-liquid phase transition appearing to play an important role in non-human mammalian oocyte spindle pole assembly. Perhaps the absence of ZYG-9 or TAC-1, even if they do not obviously appear to form condensates themselves during *C. elegans* oocyte spindle assembly, nevertheless can alter the properties of condensates that are involved in pole coalescence, such that they are less stable and more prone to splitting or nascent assembly, and to rapid growth. Higher spatial and temporal resolution imaging of ZYG-9 and TAC-1, and other factors, during oocyte meiotic spindle assembly may provide further insight.

## ZYG-9 and TAC-1 promote microtubule instability during oocyte meiotic cell division

The negative regulation of overall microtubule levels by ZYG-9 and TAC-1 in *C. elegans* oocytes is in stark contrast to its subsequent role in promoting microtubule stability during the first embryonic mitosis, which immediately follows the completion of oocyte meiosis II. Furthermore, most studies in other model systems, both in vivo and in vitro, have focused on XMAP215 and TACC roles as microtubule stabilizing factors [24]. For example, budding yeast XMAP215/Stu2p has been shown to synergize with gamma-tubulin to promote microtubule nucleation and polymerization [47,48], and XMAP215 orthologs are often described as microtubule polymerases [24,35,51]. While it is possible that ZYG-9 and TAC-1 also promote microtubule stability during oocyte meiotic cell division, the substantial increase in both spindle and cortex-associated microtubule levels after strong reduction of either ZYG-9 or TAC-1 clearly indicates a predominantly negative role during *C. elegans* oocyte meiotic cell division.

Whether XMAP215 and TACC family members act exclusively positively or negatively to regulate microtubule stability in any one setting, or instead generally use a balance of positive and negative regulation, warrants further investigation. Though less extensively studied, XMAP215 family members also destabilize microtubules in some contexts. Studies of budding yeast XMAP215/Stu2p both in vitro and in vivo indicate that it can limit microtubule length and decrease the rate of microtubule catastrophe [31,32], while studies of interphase microtubules in *Drosophila* S2 cells suggest that XMAP215/Mini-spindles negatively regulates pauses in microtubule growth, although this role may be independent of D-TACC [33]. Finally, in vitro studies using purified proteins or *Xenopus* extracts have shown that XMAP215 can promote both microtubule stability [49,51,52] and microtubule instability [34]. *C. elegans* oocyte meiotic and mitotic spindle assembly provide particularly appealing models for further investigating the role of XMAP215 and TACC family members as negative regulators of microtubule stability, and for how they can transition from being overall negative to overall positive regulators.

## Conservation of oocyte spindle assembly mechanisms across animal phyla

Studies of XMAP215 and TACC orthologs during meiotic cell division in *Drosophila* and mammalian oocytes indicate that their roles during acentrosomal spindle assembly may not be fully conserved, as the defects in some cases resemble those we have observed in *C. elegans*, and in other cases they appear distinct. In *Drosophila* oocytes, XMAP215/Mini-spindles and D-TACC also are enriched at spindle poles, and loss of either results in many oocytes assembling tripolar spindles that are remarkably similar to the tripolar spindles we frequently observed in *zyg-9* and *tac-1* mutants. However, Mini-spindles and D-TACC require the minus-end directed kinesin 14 family member Ncd for their localization to spindle poles [53], and Ncd mutant oocytes also often assemble tripolar spindles [54]. The localization of ZYG-9 and TAC-1 in *klp-15/16* mutant oocytes has not been reported, but loss of the kinesin 14 family

members KLP-15/16 in *C. elegans* results in mutant oocytes with a pole coalescence phenotype that differs substantially from the *zyg-9* and *tac-1* mutant phenotype [21], suggesting that ZYG-9 and TAC-1 may not function so directly with kinesin 14 family members as do their *Drosophila* orthologs.

In an extensive survey of mammalian oocyte spindle proteins, XMAP215/chTOG and TACC3 and several other spindle pole proteins were shown to undergo phase transitions and be present in spindle-associated condensates [22]. However, as noted earlier, GFP fusions to ZYG-9 and TAC-1 do not obviously appear to form condensates in *C*. elegans oocytes but are more diffusely localized throughout the spindle with some enrichment at the poles [21]. Moreover, in contrast to the elevated microtubule levels and the extensive defects in spindle bipolarity observed in *zyg-9* and *tac-1* mutant oocytes, bipolar spindles assembled with significantly reduced spindle microtubule levels after depletion of chTOG or TACC3 from mouse oocytes, and no defects in pole assembly or spindle bipolarity were reported [22]. Nevertheless, it is intriguing that in mouse oocytes depleted of endogenous TACC3, but expressing a TACC3 construct lacking the N-terminal residues required for phase separation, oocyte spindle poles were destabilized [22]. Finally, normal human oocytes are exceptional in exhibiting dramatic pole instability and chromosome separation defects relative to other mammalian oocytes, due at least in part to loss of expression of KIFC1/kinesin 14 [55]. In *C. elegans*, loss of KLP-15/16/ kinesin 14 results in failed pole coalescence but not in pole stability defects like those observed in *zyg-9* and *tac-1* mutant oocytes [21] and in normal human oocytes [55]. There appears to be a perplexing mix of both substantial similarities and substantial differences in the requirements for oocyte meiotic spindle assembly in different species. Higher spatial and temporal resolution studies of pole coalescence and spindle assembly dynamics in live control and mutant oocytes likely will reveal further conservation and divergence of mechanism.

## Temporal dissection of gene requirements

Fast-acting, temperature-sensitive alleles are powerful tools for high-resolution temporal dissection of gene requirements. They are particularly useful for investigating the relatively rapid oocyte meiotic and early embryo mitotic cell divisions in *C. elegans*. Coupled with devices that can control and rapidly change sample temperature during live imaging, fast-acting TS alleles have provided insights into both mitotic [39,56,57] and meiotic [8,11,40,58] cell division in *C. elegans*. However, while large-scale screens have identified thousands of temperature-sensitive, embryonic-lethal *C. elegans* mutants, relatively few have defects in early embryonic cell division [1]. Indeed, temperature-sensitive alleles have been identified in only a few hundred of the roughly 2000 *C. elegans* genes known to be essential [1], and only about half of the alleles in a collection of TS mutants with early embryonic cell division defects were found to be fast-acting [38]. While still valuable, slower-acting TS alleles require turnover of the encoded proteins and are therefore generally not useful for temporally dissecting gene requirements during the rapid early embryonic cell divisions in *C. elegans*.

The recent development of CRISPR-mediated degron tagging of endogenous loci for temporally controlled, auxin-mediated protein degradation now provides a new and powerful tool for engineering conditional loss of function alleles throughout the *C. elegans* genome [36,59]. Degron tagging also provides impressive temporal resolution, with tagged proteins in some cases substantially knocked down after only 20 to 30 minutes of exposure to auxin [36,60], and in isolated early *C. elegans* zygotes after only a few minutes of auxin exposure [37]. Fast-acting TS alleles do provide some advantages. In some cases, they may reduce gene function even more rapidly, within seconds, and they can often be used for complementary temperature up and down shift experiments to define temperature-sensitive periods. Moreover, with TS alleles

one can more readily modulate the extent to which gene function is reduced by growth at intermediate temperatures, for example to sensitize strains for modifier screens [61]. In contrast, the utility of degron tagging is sometimes limited by the tag promoting some degradation even in the absence of auxin treatment [62], and not all proteins can tolerate even the small degron tag. Nevertheless, degron tagging now makes conditional gene knockdown much more widely available for more precisely defining *C. elegans* gene requirements. With further advances in the increasingly powerful ability to predict protein structure based on primary amino acid sequence, perhaps it will become feasible to accurately predict missense mutations that are likely to generate fast-acting, temperature-sensitive mutations, further adding to our ability to temporally manipulate gene function in model organisms.

## Materials and methods

### *C. elegans* strains and maintenance

*C. elegans* strains used in this study are listed in S1 Table. Temperature-sensitive and control strains were constructed and maintained at 15˚C using previously described culturing methods [63].

The *zyg-9(or1984ts)* allele was isolated in a chemical mutagenesis screen [38,64], but has not been published previously. The *zyg-9* mis-sense mutation (D69G, GAT to GGT) in *or1984ts* was the only DNA sequence difference identified by Sanger DNA sequencing of PCR amplified fragments that spanned all exons and introns, with complementation tests to verify gene identity as summarized in Table 1.

The complex *tac-1(ok3305)* deletion/insertion allele (see WormBase) was obtained from the CGC. To determine the phenotype of homozygous *ok3305* worms, 170 progeny from a single PCR-confirmed *ok3305*/+ hermaphrodite brood cultured at 20˚C were singled out onto plates as L4s and kept at 20˚C for 48 hours, at which time their broods were scored for viability. 128 (75.3%) of the progeny produced viable broods; 42 (24.7%) of the progeny produced all dead embryos. In a separate brood analysis, 20 progeny from one PCR-confirmed *ok3305*/+ hermaphrodite were singled onto plates and allowed to produced broods; 5 produced all dead embryos and PCR detected only the deleted allele (*ok3305/ok3305*); 9 produced viable broods and PCR detected both alleles (*ok3305*/+); 6 produced viable broods and PCR detected only the WT allele (+/+), also consistent with Mendelian segregation. We conclude that the null phenotype for *tac-1(ok3305)* is recessive, maternal-effect embryonic lethality.

To determine the phenotype of homozygous *zyg-9(or1985)* worms, 189 progeny from two PCR-confirmed *or1985*/+ hermaphrodite broods cultured at 20˚C were singled out as L4s and kept at 20˚C for 48 hours. 156 (82.5%) of the progeny produced viable broods, 33 (17.5%) of the progeny produced all dead embryos. In a further test, 20 progeny from one PCR-confirmed *or1985*/+ hermaphrodite brood cultured at 20˚C were singled out onto plates as L4s and kept at 20˚C for 48 hours, at which time their broods were scored for viability. 6 produced viable broods and PCR detected only the WT band (+/+), 11 produced viable broods and PCR detected both bands (*or1985*/+), and 3 produced all dead embryos and PCR detected only the mutant band (*or1985/or1985*). We conclude that the null phenotype for *zyg-9(or1985)* is recessive, maternal-effect embryonic-lethality, although there may be a delay in larval development, or low penetrance larval lethality, that we missed, given that less than 25% of the progeny from heterozygous parents were homozygous mutant and the singled out L4s were from early in the broods of the two PCR-confirmed *or1985*/+ parents.

To determine if the embryonic lethality observed in the broods of *zyg-9(or1985)*/+ and *tac-1(ok3305)*/+ hermaphrodites (Table 1) is due to a temperature-sensitive haploinsufficiency, we analyzed broods when both deletions alleles were in trans to marked balancers and still

observed Mendelian (1:2:1) segregation of all genotypes and phenotypes in the progeny that survived to adulthood.

We used PCR and Sanger DNA sequencing to verify the presence of the TS *tac-1* and *zyg-9* mis-sense mutations in all GFP::TBB-2 and GFP::ASPM-1 strains, to ensure that all strains used for identifying *tac-1* and *zyg-9* requirements were of the correct genotype.

## CRISPR

The appropriate sgRNA and PAM sites for *zyg-9* were selected using the website http://crispor. tefor.net/. The *zyg-9* guide RNA sequences and repair oligos for both degron tagged *zyg-9* and *zyg-9* knockout are listed below. The injection mixture of *zyg-9* sgRNA(s), repair oligo (IDT), co-CRISPR marker *unc-58* repair oligo [65] (IDT), *unc-58* crRNA [65] (IDT), trRNA (IDT) and Cas9-NLS nucleases (IDT) were injected into wild-type young adults. The F1 progeny of the injected animals were first selected for the Unc phenotype then screened by PCR.

| | sgRNA(s) | Repair oligo |
|---|---|---|
| *zyg-9(or1968): degron::zyg-9* | CAATTGGGATTATCTGGACG | AAACTGTCATTTTTCAGATAATGCCTAAAGATC CAGCCAAACCTCCGGCCAAGGCACAAGTTGTGG GATGGCCACCGGTGAGATCATACCGGAAGAACG TGATGGTTTCCTGCCAAAAATCAAGCGGTGGCC CGGAGGCGGCGGCGTTCGTGAAGTCCAACTGGG ACTACCTCGATGAGGTGGATATCCTTCCCAAACTT |
| *zyg-9(or1985): zyg-9* knockout | CAATTGGGATTATCTGGACG and GGCGAACATTCGTCATACAC | GTAGTAAACTGTCATTTTTCAGATAATGTCCAATT GAGATTAACTGGCACTGGACTGAGTCTAAAATC ATTGATAATTTTATCCCTGCCCCTTAAAGT |

## Image acquisition

All imaging was performed using a Leica DMi8 microscope outfitted with a spinning disk confocal unit–CSU-W1 (Yokogawa) with Borealis (Andor), dual iXon Ultra 897 (Andor) cameras, and a 100x HCX PL APO 1.4–0.70NA oil objective lens (Leica). With the 100x objective, we measured the pixel scale to be 0.1470 +/- 0.0005 μm, versus the theoretical pixel scale of 0.160 μm from the camera's 16 x 16 μm pixels, indicating a small amount of empty magnification within the system. Metamorph (Molecular Devices) imaging software was used for controlling image acquisition. The 488nm and 561nm channels were imaged simultaneously every 10s with 1μm Z-spacing for a total stack size of 20μm.

*In utero* live imaging of oocytes was accomplished by mounting young adult worms with single row of embryos to a 5% agarose pad on a 24x40mm glass coverslip with 1.5 μl each of 15°C M9 buffer and 0.1 μm polystyrene microspheres (Polysciences Inc.), and were gently covered with a 18x18mm coverslip. Samples were then transferred to a fluidic temperature-controlled CherryTemp chip (CherryBiotech) that enabled imaging at precisely the indicated temperature and rapid temperature-shift experiments. We monitor the temperature of agar pads using a probe attached to a digital thermometer to verify the accuracy of our device, and the CherryTemp device monitors the temperature of the objective and adjusts device flow to accommodate for increased objective temperatures.

## Temperature-shift experiments

For imaging control and TS mutant oocytes kept at the permissive or restrictive temperature throughout oocyte meiosis I, worm strains were maintained at the permissive temperature of 15°C until young adults were mounted on agar pads in a 15°C cold room and maintained at

15˚C on a metal block until transfer to the CherryTemp chip (see above) pre-set at 15˚C or 26˚C and mounting on the spinning disk microscope. Imaging was begun upon identification of a pre-nuclear envelope breakdown oocyte. For multipolar upshift and downshift experiments, worms were mounted as described above onto agar pads kept at 15˚C or 26˚C using the CherryTemp device, and temperature-shifts were done after ovulation and preceding spindle bipolarity and fell within a range of 2.7 to 6.8 minutes after cage onset. For metaphase upshift experiments, temperature-shifts were performed after spindles achieved bipolarity and aligned chromosomes and preceding spindle shortening and fell into a range of 8.7 to 18 minutes after cage onset. Upshifted oocytes that did not meet these criteria or that subsequently arrested were excluded from analysis. When analyzing the time intervals for multipolar and metaphase upshifts relative to spindle shortening, both control and TS mutant oocytes at 26˚C progressed through meiosis I much faster than at 15˚C, as expected (S13 Fig). Similarly, oocytes upshifted at the multipolar stage reached anaphase onset more quickly than oocytes upshifted at metaphase (S13 Fig). To describe our temperature upshift and downshift experiments in all figures, we define t = 0 as the first frame prior to the appearance of a cage structure.

## Auxin treatment

Young adult degron::zyg-9 worms cultured on NGM plates were moved onto 1mM auxin-containing NGM plate 2–6 hours before mounting worms for live imaging. Control worms were the same strain grown on NGM plates without auxin.

## Image processing and analysis

General image processing including merging and cropping of red/green channels and z-projected images was done through FIJI [66]. All time lapse images shown in the figures are Maximum Intensity Projections of all focal planes unless otherwise noted. To account for differences in movie signal quality inherent to *in utero* live cell imaging, the intensity scales for montages in all figures were set individually to give the clearest depiction of the spindle and chromosomes; all pixel intensity based quantification was done using the raw unadjusted pixel values. The time lapse images in Figs 6 and S8 were obtained after rotation of 3-dimensional images using Imaris, with Imaris snapshots (roughly equivalent to Maximum Intensity Projections) used to acquire images of the rotated stacks. The montage frames for each stage were chosen as follows: "NEBD" is the timepoint immediately preceding the appearance of microtubule bundles forming the cage structure and serves as t = 0; "Cage" is the timepoint that best displays the spherical cage-like network of microtubules that forms around the chromosomes; "Multipolar" is the timepoint that best displays the coalescing microtubule network after the cage stage and ovulation but preceding spindle bipolarity; "Metaphase" is the timepoint that best displays the bipolar spindle with chromosomes aligned at the metaphase plate and preceding spindle shortening; as metaphase is unidentifiable in TS mutant spindles at 26˚C, the "Metaphase" frames in those montages are instead from the timepoint best displaying abnormal spindle morphology before the start of spindle shortening; "Anaphase" is a timepoint during anaphase A showing the shortened spindle and preceding chromosome separation; "End of MI" is a timepoint after chromosome separation that best displays the number of chromosome sets separated during anaphase. For montages of temperature-shift experiments, the frame at the point of upshift/downshift is always shown. For montages of meiosis II, frames for each stage were chosen as in meiosis I except "PBE I" is the timepoint where meiosis I chromosome separation ends and serves as t = 0.

To quantify the ratio of mCherry::H2B signal of chromosome sets in two-way anaphases, a sum projection of the red channel was created in FIJI and then background subtracted using a 50 pixel rolling ball radius. A region of interest was drawn around each chromosome mass at the end of chromosome separation and a ratio was made of the mean sum intensity.

Normalized microtubule pixel intensity was quantified in FIJI as described previously [21]. For normalized microtubule intensity quantification by meiotic stage (Fig 3A), measurements from three consecutive time points were taken at each stage per oocyte and reported as an average. The stages are defined as follows: the "Cage" is when GFP::TBB-2 marked microtubules formed a spherical cage-like network around the chromosomes; "Multipolar" is the time-point midway between the Cage and Metaphase stages; "Metaphase" is 30 seconds before the onset of spindle shortening; "Anaphase" is the halfway point from spindle shortening onset to the start of chromosome separation; "Telophase" is midway through chromosome separation. For combined normalized microtubule intensities (Fig 3B), the microtubule intensity values for all meiotic stages as quantified for Fig 3A were taken from each oocyte and grouped by genotype.

Three-dimensional projection and rotation movies were made using Imaris software (Bitplane) and were used to analyze cage structures (Fig 2K), assess pole splitting (Table 1), score spindle pole numbers (Figs 1I, 2I, 5L and 8I), and analyze chromosome separation outcomes (Figs 1J, 2J, 5N and 8J) and polar body extrusion (Fig 7G).

A cage structure was scored as defective when GFP::TBB-2 marked microtubule bundles passed through the interior of the nucleus instead of maintaining a roughly spherical cage around the chromosomes (Figs 2K and S7, S9–S11 Movies).

For assessing pole splitting (Table 1), GFP::ASPM-1 marked foci were monitored from coalescence until spindle shortening and a splitting event was defined as when a single focus fissions into two or more separate foci (S12 Movie).

To quantify spindle pole numbers at the time of spindle shortening, we counted the number of major GFP::TBB-2 microtubule asters or GFP::ASPM-1 pole foci (Figs 1I, 2I, 5L and 8I).

To evaluate anaphase outcomes (Figs 1J, 2J, 5N and 8J), we recorded the number of mCherry::H2B marked chromosome sets in each oocyte at the end of meiosis I or II.

To quantify GFP::ASPM-1 foci volumes and intensities (Fig 4M–4P), we used the Spots creation tool in Imaris (S25 Movie). In the Spots creation wizard, we enabled "different spot sizes (region growing)" to capture foci across a range of sizes. The source channel for detection was set to "Green Channel" and the estimated XY diameter for Spot detection was 1.47μm. The background subtraction (local contrast) method was used to threshold foci, and the threshold value was adjusted manually until Spots appeared over the GFP::ASPM-1 foci of the forming oocyte spindle. During Spot detection, the region growing diameter was measured from the region volume. To eliminate off-targets, we filtered the Spot creation to exclude Spots further than 10μm from the chromosomes marked by mCherry::H2B and then manually deleted remaining off-target Spots. Spot volume and intensity data from each movie were then exported to Microsoft Excel (Microsoft), the intensity data were normalized to a 0–1 scale by subtracting the data by its minimum value and then dividing by its maximum value.

Spindle rotation defects were quantified in two ways. First (Fig 7E), we scored the number of mCherry::H2B marked chromosomes merged with the cortex at the onset of chromosome separation as described previously [42]. Second (Fig 7F), we recorded the spindle angle relative to the cortex twenty seconds prior to the start of chromosome separation using FIJI [42]. Spindles were only analyzed if both poles were present in the same focal plane, indicating a flat orientation, and we recorded the intersecting angle of a line bisecting the spindle in the pole-to-pole axis and a second line tangential to the cortex.

Meiosis I polar body extrusion success (Fig 7G) was evaluated based on whether oocytes extruded any chromosomes marked by mCherry::H2B into a polar body that remained extruded until meiosis II spindle assembly began, as described previously [46].

## Statistics

P-values comparing distributions for all scatter plots were calculated using the Mann–Whitney U-test. P-values comparing slopes were calculated using a two-tailed t-test. Statistical analysis was performed using Microsoft Excel (Microsoft) and Prism 9 (GraphPad Software) and graphs were made in Prism 9. Raw data for all statistical analyses are provided in Excel spreadsheets (see S1–S4 Data).

## Supporting information

**S1 Table.** *C. elegans* **strains used in this study.**
(TIF)

**S1 Fig. Sequence changes in** *tac-1* **and** *zyg-9* **mutant alleles.** Schematic diagrams indicating locations of the mis-sense mutations in TS alleles (red vertical lines), the *tac-1(ok3305)* and *zyg-9(or1985)* deletion endpoints (horizontal red line), the TAC-1 coiled-coil TACC domain in light grey, and the ZYG-9 TOG domains in dark grey, with amino acid boundaries of domains indicated. The *tac-1(ok3305)* allele is a complex rearrangement with a large deletion and small 23 base pair insertion 5' to the deletion at the junction of exon 1 and intron 1 that introduces four extra amino acids to the C-terminus of the exon 1 encoded amino acids (see WormBase).
(TIF)

**S2 Fig. Timing of temperature shifts in control and TS mutant oocytes.** (A-D) All upshift and downshift timepoints for multipolar and metaphase temperature-shift experiments over-laid onto the timing of meiotic events at 15˚C for each allele. Note that the multipolar down-shifts are shifted to slightly earlier time points due to the more rapid development at 26˚C prior to the downshifts. Error bars and values are mean ± the range (S1 Data). (E) Tables displaying the ranges of multipolar and metaphase temperature-shift timepoints relative to cage onset at 15˚C for each allele.
(TIF)

**S3 Fig. Meiosis I spindle assembly dynamics in control and TS mutant oocytes at the permissive and restrictive temperatures.** (A-H) Time-lapse maximum intensity projection images during meiosis I in live control and TS mutant oocytes expressing GFP::TBB-2 and mCherry::H2B to mark microtubules and chromosomes, at 15˚C (A-D) and at 26˚C (E-H). In this and in all subsequent meiosis I time-lapse image series, t = 0 is labeled NEBD and is the timepoint immediately preceding the appearance of microtubule bundles forming the cage structure. See the Materials and Methods for a description of the assembly stages and frame selections in this and subsequent supplemental figures. Scale bars = 5 μm.
(TIF)

**S4 Fig. Degron and RNAi knockdowns of ZYG-9 and TAC-1.** (A-D) Time-lapse maximum intensity projection images of live control (-Auxin) and auxin-induced knockdown (+Auxin) of degron-tagged endogenous *zyg-9* locus at indicated stages. mCherry::H2b in magenta; GFP::ASPM-1 in teal. (E) Bar graphs quantifying the spindle polarity and chromosome separation defects in control and auxin treated oocytes. (F) RNAi knockdowns using previously reported conditions and endogenous GFP fusion strains [21]. As also reported by others [44], all GFP::

TAC-1 signal is lost from spindle microtubules and poles after *zyg-9* RNAi, while GFP::ZYG-9 is lost from poles but not spindle microtubules after *tac-1(RNAi)*. Scale bars = 5 μm.
(TIF)

**S5 Fig. Meiosis I spindle assembly dynamics in control and TS mutant oocytes subjected to prometaphase temperature shifts.** (A-G) Time-lapse maximum intensity projection images of live control and TS mutant oocytes upshifted to 26°C (A-D) or downshifted to 15°C (E-G) during the multipolar stage, in oocytes expressing GFP::TBB-2 and mCherry::H2B. Oocytes depicted in G have cage structure defects (for an example, see S9 Movie). Scale bars = 5 μm.
(TIF)

**S6 Fig. Meiosis I pole assembly dynamics in control and TS mutant oocytes at the permissive and restrictive temperatures.** (A-F) Time-lapse maximum intensity projection images of live control and TS mutant oocytes expressing GFP::ASPM-1 and mCherry::H2B to mark spindle poles and chromosomes, at 15°C (A-C), at 26°C (D-F). Scale bars = 5 μm.
(TIF)

**S7 Fig. Meiosis I pole assembly dynamics in control and TS mutant oocytes subjected to prometaphase temperature shifts.** (A-F) Time-lapse maximum intensity projection images of live control and TS mutant oocytes expressing GFP::ASPM-1 and mCherry::H2B to mark spindle poles and chromosomes, and upshifted to 26°C (A-C) or downshifted to 15°C (D-F) during the multipolar stage. Montage frames highlight pole coalescence dynamics during the multipolar stage through to the onset of spindle shortening. Scale bars = 5 μm.
(TIF)

**S8 Fig. Meiosis I pole growth dynamics in control and *tac-1(or455ts)* oocytes subjected to prometaphase temperature shifts.** (M-P) Quantification of control and *tac-1(or455ts)* GFP::ASPM-1 foci volume and intensity (see Materials and Methods) pre- and post-multipolar upshift (A, B) and downshift (C,D). Slopes were compared using a two-tailed t-test to calculate P-values (S4 Data). (E) Table showing the time elapsed pre- and post-multipolar upshift (upper table) and multipolar downshifts (lower table) for control and TS mutants. *, P <0.05; ****, P <0.0001.
(TIF)

**S9 Fig. Meiosis I pole and chromosome dynamics in control and TS mutant oocytes subjected to metaphase temperature upshifts.** (A-H) Time-lapse maximum intensity projection images of live control and TS mutant oocytes upshifted at metaphase and expressing either GFP::ASPM-1 and mCherry::H2B (A-D) or GFP::TBB-2 and mCherry::H2B (E-H). Montage frames highlight defects following metaphase upshift through to the end of meiosis I. White outlined arrowheads denote ectopic spindle poles and solid white arrowheads indicate chromosome congression errors. The montage in B (top row) is also depicted in Fig 7B. Scale bars = 5 μm.
(TIF)

**S10 Fig. Imaris snapshots of meiosis I pole and chromosome dynamics in control and TS mutant oocytes subjected to metaphase temperature upshifts.** (A-H) Imaris rotated and snapshot projected time-lapse images (see Materials and Methods) of live TS mutant oocytes upshifted at metaphase expressing GFP::ASPM-1 and mCherry::H2B (A-D) or GFP::TBB-2 and mCherry::H2B (E-H). Montage frames highlight defects following metaphase upshift through to spindle shortening. White outlined arrowheads denote ectopic spindle poles; solid white arrowheads indicate chromosome congression errors. Imaris montages of A-H are of the same oocytes shown in the top rows of S7 Fig as maximum intensity projection montages

B-H. Scale bars = 5 μm.
(TIF)

**S11 Fig. Meiosis I spindle rotation and polar body extrusion in control and TS mutant oocytes subjected to metaphase temperature upshifts.** (A-H) Time-lapse maximum intensity projection images of live control and TS mutant oocytes expressing GFP::ASPM-1 and mCherry::H2B or GFP::TBB-2 and mCherry::H2B upshifted at meiosis I metaphase. Dashed lines depict the oocyte cortex. Montages with a white circle in the last frame indicate failed polar body extrusion. The montage in B (top row) is also depicted in Fig 5B. Montages in E-H are also depicted in S7E Fig bottom row, Fig 5I, S7G Fig middle row, and S7D Fig bottom row, respectively. (I) Table showing the correlation between spindle rotation defects and polar body extrusion failure. Only oocytes in which both spindle rotation and polar body extrusion could be scored are included. Scale bars = 5 μm.
(TIF)

**S12 Fig. Meiosis II spindle pole assembly dynamics in control and TS mutant oocytes at the permissive and restrictive temperatures.** (A-H) Time-lapse maximum intensity projection images during meiosis II of live control and TS mutant oocytes expressing GFP::ASPM-1 and mCherry::H2B at 15˚C (A-D) and at 26˚C (E-H). t = 0 is the timepoint when meiosis I chromosome separation ends. Scale bars = 5 μm.
(TIF)

**S13 Fig. Temperature effects on meiosis I cell cycle timing in control and TS mutant oocytes.** (A-D) The timing of meiotic events in oocytes maintained at 15˚C or 26˚C, and in oocytes that underwent multipolar and metaphase upshifts, in control and TS mutant oocytes. Error bars and values are mean ± the range. (E) Table showing the mean time to spindle shortening in control and TS mutant oocytes for each temperature condition (S1 Data). Spindle bipolarity and chromosome alignment is rarely achieved in multipolar upshifted or TS mutant oocytes at 26˚C and so is not scored; ovulation does occur in TS mutants at 26˚C but was not scored.
(TIF)

**S1 Movie. Microtubule dynamics in a control oocyte maintained at 26˚C throughout meiosis I.** *In utero* time-lapse spinning disk confocal Imaris movie of meiosis I in a control oocyte at 26˚C expressing GFP::TBB-2 and mCherry::H2B. For this and all supplemental movies, Z-stacks were collected every 10 seconds; frame rate is 10 frames per second. This oocyte is shown in Fig 1E.
(MP4)

**S2 Movie. Microtubule dynamics in a *zyg-9(or1984ts)* oocyte maintained at 26˚C throughout meiosis I.** *In utero* time-lapse spinning disk confocal Imaris movie of meiosis I in a *zyg-9 (or1984ts)* oocyte at 26˚C expressing GFP::TBB-2 and mCherry::H2B. Frame rate is 10 frames per second. This oocyte is shown in Fig 1F.
(MP4)

**S3 Movie. Microtubule dynamics in a multipolar stage upshifted control oocyte.** *In utero* time-lapse spinning disk confocal Imaris movie of meiosis I in a control oocyte upshifted at the multipolar stage expressing GFP::TBB-2 and mCherry::H2B. Upshift timepoint is indicated by an orange sphere. Frame rate is 10 frames per second. This oocyte is shown in S3A Fig.
(MP4)

**S4 Movie. Microtubule dynamics in a multipolar stage upshifted *zyg-9(or1984ts)* oocyte.** *In utero* time-lapse spinning disk confocal Imaris movie of meiosis I in a *zyg-9(or1984ts)* oocyte

upshifted at the multipolar stage expressing GFP::TBB-2 and mCherry::H2B. Upshift time-point is indicated by an orange sphere. Frame rate is 10 frames per second. This oocyte is shown in Fig 2B.
(MP4)

**S5 Movie. Microtubule dynamics in a multipolar stage upshifted *zyg-9(or634ts)* oocyte.** *In utero* time-lapse spinning disk confocal Imaris movie of meiosis I in a *zyg-9(or634ts)* oocyte upshifted at the multipolar stage expressing GFP::TBB-2 and mCherry::H2B. Upshift time-point is indicated by an orange sphere. Frame rate is 10 frames per second. This oocyte is shown in Fig 2C.
(MP4)

**S6 Movie. Microtubule dynamics in a multipolar stage upshifted *tac-1(or455ts)* oocyte.** *In utero* time-lapse spinning disk confocal Imaris movie of meiosis I in a *tac-1(or455ts)* oocyte upshifted at the multipolar stage expressing GFP::TBB-2 and mCherry::H2B. Upshift time-point is indicated by an orange sphere. Frame rate is 10 frames per second. This oocyte is shown in Fig 2D.
(MP4)

**S7 Movie. Rotation of a control oocyte cage structure.** *In utero* spinning disk confocal Imaris rotation movie of a normal cage structure restricted to the periphery in a control oocyte at 26˚C expressing GFP::TBB-2 and mCherry::H2B.
(MP4)

**S8 Movie. Spindle assembly defects in a multipolar stage upshifted *tac-1(or455ts)* oocyte with a normal cage structure.** *In utero* time-lapse spinning disk confocal Imaris movie of meiosis I in a *tac-1(or455ts)* oocyte upshifted at the multipolar stage expressing GFP::TBB-2 and mCherry::H2B. Rotation displays normal cage structure. Upshift timepoint is indicated by an orange sphere. Frame rate is 10 frames per second. This oocyte is shown in S3D Fig (top row).
(MP4)

**S9 Movie. Rescue of bipolar spindle assembly in a multipolar stage downshifted *tac-1 (or455ts)* oocyte with a cage structure defect.** *In utero* time-lapse spinning disk confocal Imaris movie of meiosis I in a *tac-1(or455ts)* oocyte downshifted at the multipolar stage expressing GFP::TBB-2 and mCherry::H2B. Rotation displays abnormal cage structure with microtubule bundles passing in between chromosomes through the internal volume. Downshift timepoint is indicated by a green sphere. Frame rate is 10 frames per second. This oocyte is shown in Fig 2H.
(MP4)

**S10 Movie. Rescue of bipolar spindle assembly in a multipolar stage downshifted *tac-1 (or455ts)* oocyte with a cage structure defect.** *In utero* time-lapse spinning disk confocal Imaris movie of meiosis I in a *tac-1(or455ts)* oocyte downshifted at the multipolar stage expressing GFP::TBB-2 and mCherry::H2B. Rotation displays abnormal cage structure with microtubule bundles passing in between chromosomes through the internal volume. Downshift timepoint is indicated by a green sphere. Frame rate is 10 frames per second. This oocyte is shown in Fig 2G.
(MP4)

**S11 Movie. Rescue of bipolar spindle assembly in a multipolar stage downshifted *tac-1 (or455ts)* oocyte with a cage structure defect.** *In utero* time-lapse spinning disk confocal Imaris movie of meiosis I in a *tac-1(or455ts)* oocyte downshifted at the multipolar stage expressing

GFP::TBB-2 and mCherry::H2B. Rotation displays abnormal cage structure with microtubule bundles passing in between chromosomes through the internal volume. Downshift timepoint is indicated by a green sphere. Frame rate is 10 frames per second. This oocyte is shown in S5G Fig montage 2.
(MP4)

**S12 Movie. GFP::ASPM-1 pole splitting event in a multipolar stage upshifted *tac-1 (or455ts)* oocyte.** Excerpt of an *in utero* time-lapse spinning disk confocal Imaris movie of a *tac-1(or455ts)* oocyte upshifted at the multipolar stage expressing GFP::ASPM-1 and mCherry::H2B. Upshift timepoint is indicated by an orange sphere. Frame rate is 3 frames per second.
(MP4)

**S13 Movie. Pole assembly dynamics in a multipolar stage upshifted control oocyte.** *In utero* time-lapse spinning disk confocal Imaris movie of NEBD to spindle shortening onset in a control oocyte upshifted at the multipolar stage expressing GFP::ASPM-1 and mCherry::H2B. Upshift timepoint is indicated by an orange sphere. Frame rate is 5 frames per second. This oocyte is shown in S5A Fig (top row).
(MP4)

**S14 Movie. Pole assembly dynamics in a multipolar stage upshifted *zyg-9(or1984ts)* oocyte.** *In utero* time-lapse spinning disk confocal Imaris movie of NEBD to spindle shortening onset in a *zyg-9(or1984ts)* oocyte upshifted at the multipolar stage expressing GFP::ASPM-1 and mCherry::H2B. Upshift timepoint is indicated by an orange sphere. Frame rate is 5 frames per second. This oocyte is shown in Fig 4H.
(MP4)

**S15 Movie. Pole assembly dynamics in a multipolar stage upshifted *tac-1(or455ts)* oocyte.** *In utero* time-lapse spinning disk confocal Imaris movie of NEBD to spindle shortening onset in a *tac-1(or455ts)* oocyte upshifted at the multipolar stage expressing GFP::ASPM-1 and mCherry::H2B. Upshift timepoint is indicated by an orange sphere. Frame rate is 5 frames per second. This oocyte is shown in Fig 4I.
(MP4)

**S16 Movie. Pole assembly dynamics in a multipolar stage downshifted control oocyte.** *In utero* time-lapse spinning disk confocal Imaris movie of NEBD to spindle shortening onset in a control oocyte downshifted at the multipolar stage expressing GFP::ASPM-1 and mCherry:: H2B. Downshift timepoint is indicated by a green sphere. Frame rate is 5 frames per second. This oocyte is shown in Fig 4J.
(MP4)

**S17 Movie. Pole assembly dynamics in a multipolar stage downshifted *zyg-9(or1984ts)* oocyte.** *In utero* time-lapse spinning disk confocal Imaris movie of NEBD to spindle shortening onset in a *zyg-9(or1984ts)* oocyte downshifted at the multipolar stage expressing GFP:: ASPM-1 and mCherry::H2B. Downshift timepoint is indicated by a green sphere. Frame rate is 5 frames per second. This oocyte is shown in Fig 4K.
(MP4)

**S18 Movie. Pole assembly dynamics in a multipolar stage downshifted *tac-1(or455ts)* oocyte.** *In utero* time-lapse spinning disk confocal Imaris movie of NEBD to spindle shortening onset in a *tac-1(or455ts)* oocyte downshifted at the multipolar stage expressing GFP:: ASPM-1 and mCherry::H2B. Downshift timepoint is indicated by a green sphere. Frame rate

is 5 frames per second. This oocyte is shown in Fig 4L.
(MP4)

**S19 Movie. Ectopic pole formation in a metaphase stage upshifted** *zyg-9(or634ts)* **oocyte.** *In utero* time-lapse spinning disk confocal Imaris movie of a *zyg-9(or634ts)* oocyte upshifted at metaphase expressing GFP::ASPM-1 and mCherry::H2B. Movie is from the point of upshift (indicated by an orange sphere) to the end of meiosis I. Frame rate is 5 frames per second. This oocyte is shown in Fig 5C.
(MP4)

**S20 Movie. Chromosome congression defect in a metaphase stage upshifted** *zyg-9 (or1984ts)* **oocyte.** *In utero* time-lapse spinning disk confocal Imaris movie of a *zyg-9 (or1984ts)* oocyte upshifted at metaphase expressing GFP::ASPM-1 and mCherry::H2B. Movie is from the point of upshift (indicated by an orange sphere) to the end of meiosis I. Frame rate is 5 frames per second. This oocyte is shown in S7B Fig (top row) and Fig 7B.
(MP4)

**S21 Movie. Ectopic pole associated with a poorly congressed bivalent in a metaphase stage upshifted** *zyg-9(or1984ts)* **oocyte.** *In utero* time-lapse spinning disk confocal Imaris movie of a *zyg-9(or1984ts)* oocyte upshifted at metaphase expressing GFP::ASPM-1 and mCherry::H2B. Movie is from the point of upshift (indicated by an orange sphere) to the end of meiosis I. Rotation shows the ectopic pole associated with a poorly congressed bivalent. Frame rate is 5 frames per second. This oocyte is shown in S7B Fig (bottom row).
(MP4)

**S22 Movie. Ectopic microtubule bundle associated with a poorly congressed bivalent in a metaphase stage upshifted** *tac-1(or455ts)* **oocyte.** *In utero* time-lapse spinning disk confocal Imaris movie of a *tac-1(or455ts)* oocyte upshifted at metaphase expressing GFP::TBB-2 and mCherry::H2B. Movie is from the point of upshift (indicated by an orange sphere) to the end of meiosis I. Rotation shows the ectopic microtubule bundle associated with a poorly congressed bivalent. Frame rate is 5 frames per second. This oocyte is shown in Fig 5K.
(MP4)

**S23 Movie. Spindle rotation failure in a metaphase stage upshifted** *tac-1(or455ts)* **oocyte.** *In utero* time-lapse spinning disk confocal Imaris movie of a *tac-1(or455ts)* oocyte upshifted at metaphase expressing GFP::ASPM-1 and mCherry::H2B. Movie is from the point of upshift (indicated by an orange sphere) to the end of meiosis I. Rotation shows the vertically oriented spindle at point of upshift. This oocyte is shown in Fig 7D.
(MP4)

**S24 Movie. Polar body extrusion failure in a metaphase stage upshifted** *zyg-9(or634ts)* **oocyte.** *In utero* time-lapse spinning disk confocal Imaris movie of a *zyg-9(or634ts)* oocyte upshifted at metaphase expressing GFP::TBB-2 and mCherry::H2B. Movie is from the point of upshift (indicated by an orange sphere) to the start of meiosis II. Rotation shows both chromosome sets present in the cytoplasm at meiosis II onset. Frame rate is 10 frames per second. This oocyte is shown in S7G Fig (top row) and S9C Fig (bottom row).
(MP4)

**S25 Movie. Imaris 'Spots' module capturing GFP::ASPM-1 foci in a control oocyte.** *In utero* time-lapse spinning disk confocal Imaris movie of Spots created on GFP::ASPM-1 signal in a control oocyte upshifted at the multipolar stage expressing GFP::ASPM-1 and mCherry::H2B. This movie is the same as S11 Movie except displaying Imaris Spots. Upshift timepoint is

indicated by an orange sphere. Frame rate is 5 frames per second.
(MP4)

**S1 Data. Raw data for timing of temperature shifts during meiosis I in S2 and S13 Figs.**
(XLSX)

**S2 Data. Raw data for statistics on chromosome set ratios in Figs 1 and 5.**
(XLSX)

**S3 Data. Raw data for quantification of microtubule levels in Fig 3.**
(XLSX)

**S4 Data. Raw data for quantifying GFP::ASPM-1 spots in Figs 4 and S8.**
(XLSX)

## Author Contributions

**Conceptualization:** Austin M. Harvey, Bruce Bowerman.

**Data curation:** Austin M. Harvey, Chien-Hui Chuang, Eisuke Sumiyoshi.

**Formal analysis:** Austin M. Harvey, Chien-Hui Chuang, Eisuke Sumiyoshi.

**Investigation:** Austin M. Harvey, Chien-Hui Chuang, Eisuke Sumiyoshi.

**Methodology:** Bruce Bowerman.

**Project administration:** Bruce Bowerman.

**Supervision:** Bruce Bowerman.

**Validation:** Austin M. Harvey, Chien-Hui Chuang, Eisuke Sumiyoshi, Bruce Bowerman.

**Visualization:** Austin M. Harvey, Chien-Hui Chuang, Eisuke Sumiyoshi, Bruce Bowerman.

**Writing – original draft:** Austin M. Harvey, Bruce Bowerman.

**Writing – review & editing:** Austin M. Harvey, Bruce Bowerman.

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
