## [Decision Letter · Decision Letter 0]

30 Aug 2022

Dear Dr Bowerman,

Thank you very much for submitting your Research Article entitled 'C. elegans XMAP215/ZYG-9 and TACC/TAC-1 act at multiple times throughout oocyte meiotic spindle assembly to promote both the coalescence of pole foci into a bipolar structure and the stability of coalesced poles during oocyte meiotic cell division' to PLOS Genetics.

The manuscript was fully evaluated at the editorial level and by independent peer reviewers. The reviewers appreciated the attention to an important problem, but raised some substantial concerns about the current manuscript. Based on the reviews, we will not be able to accept this version of the manuscript, but we would be willing to review a much-revised version. We cannot, of course, promise publication at that time.

In particular, please make the suggested modifications to the text and figures, and include the suggested schematics, since all of these combined will further strengthen your findings, ensure reproducibility by others in the field, and make it easier for readers to follow all the information being presented. In light of comments by Reviewer 1, we ask that you rethink some of the conclusions or at a minimum include alternative interpretations in the body of the main text (i.e. data in Figure 4). We also ask that you include all raw data with the revised manuscript (this can be uploaded as Supplemental Data). Finally, including some type of assessment of protein levels would address points raised by both reviewers.

If you decide to revise the manuscript for further consideration at PLOS Genetics, please aim to resubmit within the next 60 days, unless it will take extra time to address the concerns of the reviewers, in which case we would appreciate an expected resubmission date by email to plosgenetics@plos.org.

[LINK]

We are sorry that we cannot be more positive about your manuscript at this stage. Please do not hesitate to contact us if you have any concerns or questions.

Yours sincerely,

Mónica P. Colaiácovo

Academic Editor

PLOS Genetics

Gregory P. Copenhaver

Editor-in-Chief

PLOS Genetics

Reviewer's Responses to Questions

**Comments to the Authors:**

Reviewer #1: Previous work from the same lab has shown that ZYG-9 (homolog of a conserved microtubule-binding protein that has been studied extensively in many species) is required for C. elegans oocyte meiotic spindle poles to coalesce into exactly two poles. Having exactly two poles is important for accurate reduction of chromosome number during meiosis. Therefore this is a very significant problem that is likely relevant in other species. This manuscript includes very careful analysis of live imaging data. The work is mostly very rigorous, adds reproducibility to previous work, and adds many details to the zyg-9 phenotype. However, the work does not shed significant light on the mechanism of pole coalescence and does not really add to our understanding of what a pole is. ASPM from other species has been shown to bind microtubule minus ends but EM tomography has shown that minus ends are distributed equally throughout these spindles. Three major strengths of the manuscript are 1) it is very well written and mostly does a good job of citing relevant previous work; 2) the use of rapid temperature upshift and downshift on ts mutants to separate early and late direct functions and 3) the finding that the rate of growth of ASPM foci is faster in upshifted zyg-9 mutants. Unfortunately, there are minor weaknesses associated with two of these strengths. Unlike RNAi or degron depletions, there is no way to directly assess the extent of inactivation in a temperature shift of a ts mutant because the protein might still be present but be partially denatured. For some experiments detailed below, the conclusions are not justified because shorter temperature shifts might cause less inactivation. The increased rate of ASPM focus growth observed in zyg-9(ts) was not observed in tac-1(ts). Because the pole instability and chromosome segregation defects were observed in both zyg-9(ts) and tac-1(ts), the increased rate of focus growth cannot be the cause of the larger scale, significant phenotypes. Overall, the work is very rigorous but provides only limited new insight into how poles coalesce or how zyg-9/tac-1 function.

Detailed comments intended to improve the manuscript

Introduction

Because the introduction states that the significance of this work is showing that zyg-9/tac-1 is required to maintain spindle structure after assembly, it would be appropriate to cite Cavin-Meza 2022 who showed with a dhc-1 degron that dynein is required after spindle assembly to maintain spindle structure.

It would be appropriate to cite an example where something new was discovered with a fast acting ts mutant in C. elegans. Using metaphase-arrested embryos, Wolf et al. 2022 showed that klp-18 is required to maintain bipolarity after assembly and McNally et al. 2014 showed that MEI-1 is required to maintain chromosome alignment after congression. It would enhance the readers understanding of the significance of the current work if a paper was cited in which continuous function was NOT required and it was shown that the negative result was not due to incomplete depletion/inactivation.

Results

“These results indicate that any further defects observed after temperature upshifts are due to inactivation of the mutant protein.” This is not the correct conclusion from this result. Complementation of the phenotype with a wt transgene would support this stated conclusion. The conclusion might be that “any further defects observed after temperature upshifts correspond to the observed embryonic lethality”.

The time of temperature shift for Fig. 1 needs to be stated in the results text and in the legend given the importance given to rapid shift in the introduction. The text does not state whether the worms were shifted to 26 overnight or if they were shifted 60 seconds before nuclear envelope breakdown.

“We next assessed how effectively these TS alleles reduce gene function.” This is followed by showing phenotypes similar to the Chuang 2020 paper which did not show the extent of knockdown by Western or IF. Thus the phenotypes being similar does not show “how effectively these TS alleles reduce gene function.” Comparing to a molecular null allele would accomplish the stated goal. A possibly stronger conclusion from the similar phenotypes of a TS and RNAi is that these phenotypes are actually caused by a loss of ZYG-9.

The argument that upshifted mutants do not have a prior cage defect (with microtubules passing through the center rather than being restricted to the outside of a cage) is not supported by the cited figures 2B-D and Fig. S3. In almost all of the images (eg. 2A, 2B, 2C), there are microtubule bundles present in the middle of the cage. This is possibly because z-projections are shown. If the authors wish to argue that there are empty cages with no microtubules in the middle, they need to find a different way to display this data, possible by showing single focal planes rather than z-projections. Unless better examples are shown, readers will be suspicious of the quantification in 2I, 2K. The methods state that cages were scored for 3D rotating projections in Imaris but supplemental videos of this are not cited at the relevant location in the text.

Table I. n is quite low (both for events per embryo and total number of embryos) for making the argument that pole splitting occurs more frequently in upshifted mutants than in controls.

The data in figure 4 showing that the rate of increase in the volume and intensity of GFP::ASPM foci is greater in zyg-9 mutants after upshift is very nice. The opposite effect after downshift is even more dramatic. However, the fact that this difference was not observed in the tac-1 mutant means that this is not related to the significant phenotypes which are the same between zyg-9 and tac-1. Thus the data does not support the conclusion that “To summarize, these changes in pole dynamics after temperature upshifts and downshifts suggest that ZYG-9

and TAC-1 promote pole coalescence both by promoting pole stability and by limiting pole

growth.” The sentence explaining this in the discussion is not adequate to deal with this problem.

The relative lack of multipolar spindles and anaphase defects after late temperature upshifts is used to argue that these phenotypes are due to earlier pole stability defects in early shift experiments. However, it is equally possible that zyg-9 and tac-1 are more completely inactivated by longer temperature shift.

The spindle rotation defect and polar body extrusion defects are important to analyze but both phenotypes were partial and it is not possible to tell if this is due to partial inactivation. The defects in bipolarity in meiosis II are very weak and do not convincingly show that zyg-9/tac-1 is essential for pole coalescence in meiosis II. It could be that they are essential in meiosis II and the shorter temperature shift is causing less complete inactivation but there is no way to distinguish between these possibilities.

The data availability statement does not state where the data can be found.

Methods

Somewhere in the paper, the authors should justify the argument that the observed phenotypes are caused by the cited mutations in zyg-9 or tac-1 and are not caused by linked second site mutations. For example, if these mutations were previously complemented with a transgene, this could be stated at the beginning of the Methods. For example after “C. elegans strains used in this study are listed in Table S1.”.

The methods should state how the temperature of the worms was determined. If the authors did not directly measure the temperature of the worms and instead relied on the temperature of the cherry biotech device, at the very least, room temperature should be given. The oil-immersion objective acts as a heat sink and could significantly change the temperature at the worm. This is important for others to be able to reproduce these results.

The methods should state the pixel size in the raw images. The cited camera has 16 um pixels so that with no empty magnification, images would have 160 nm pixels but the microscope might have some empty magnification. The methods should also state the approximate laser power used. This information is important for others to reproduce these results.

Reviewer #2: Assembly of the acentrosomal spindle in C. elegans is not entirely understood. One big limitation to our understanding the roles played by specific proteins is the lack of tools to acutely deplete or inactivate them. In the current manuscript, Harvey & Bowerman address the role of the ZYG-9/TAC-1 complex during different steps of meiosis I from spindle assembly to anaphase chromosome segregation using fast-acting temperature-sensitive (ts) alleles. These proteins have mainly been studied using RNAi-mediated depletion (with the exception below). These alleles allowed the authors to dissect the requirements of these proteins with high temporal resolution.

The main findings of this manuscript are that 1) ZYG-9 and TAC-1 are required for pole coalescence into a bipolar structure by stabilising pole foci and limiting their growth rate and 2) ZYG-9 and TAC-1 maintain bipolarity during Metaphase I by suppressing ectopic poles, a process important for maintaining chromosome congression. Interestingly, their results are complementary to those posted in a very recent preprint that uses a degron-based strategy for acute ZYG-9 depletion (Mullen et al 2022). This highlights the value of complementary strategies to address meiotic events with high temporal resolution to gain a better understanding of how an oocyte assembles the spindle and how it is dynamically regulated.

Overall, the experiments are well performed and explained in detail, and the results are very clear (including the strength of each phenotype).

Comments/suggestions/questions

• It would be good for the reader to have a schematic of the different mutations within ZYG-9 and TAC-1. This is a central point, as these alleles are the key tool of the paper and having a better idea of whether the alleles operate at the protein level and/or function would hopefully help the reader better understand the penetrance of different phenotypes. Are the proteins still capable of interacting in the different mutants? I can see this was partially addressed for some mutants and their impact during mitosis in Bellanger et al, 2007.

• I also think it would be good for the reader to have a schematic of the temperature shift in relation to meiosis progression to add a visual cue for better orientation in the figure. While t=0 is explained in the methods section, it could be relevant to include these in the main text and/or Fig.

• Figures could benefit from having arrows pointing to relevant characteristics mentioned in the text.

• Are the levels of the proteins altered in the mutants? Since these proteins form a complex, analysis of their levels in the different mutants could be interesting. Either WB of endogenous proteins or maybe using fluorescently labelled ts alleles.

• For Figure 3, the text states that since spindle assembly and chromosome segregation were often defective in all three ts mutants at 26 deg, elevation of MT levels cannot account for the bipolarity and segregation defects. what is the incidence of defective bipolarity within the movies with normal MT levels?

• In Figure 5, is the resolution seems particularly low and it does not help to easily identify ectopic microtubule bundles.

**Have all data underlying the figures and results presented in the manuscript been provided?**

Reviewer #1: **No: **Means and SEM are provided in figures but I do not see a supplemental spreadsheet with all of the raw vales. The journals policy on the website says that "available upon request" is not adequate.

Reviewer #2: Yes

PLOS authors have the option to publish the peer review history of their article (what does this mean?). If published, this will include your full peer review and any attached files.

Reviewer #1: No

Reviewer #2: **Yes: **Federico Pelisch

---

## [Decision Letter · Decision Letter 1]

13 Dec 2022

Dear Dr Bowerman,

We are pleased to inform you that your manuscript entitled "C. elegans XMAP215/ZYG-9 and TACC/TAC-1 act at multiple times during oocyte meiotic spindle assembly and promote both spindle pole coalescence and stability" has been editorially accepted for publication in PLOS Genetics. Congratulations!

Please take a look at the two minor suggestions made by Reviewer #1 and incorporate these to your final version if possible. Before your submission can be formally accepted and sent to production you will need to complete our formatting changes, which you will receive in a follow up email. Please be aware that it may take several days for you to receive this email; during this time no action is required by you. Please note: the accept date on your published article will reflect the date of this provisional acceptance, but your manuscript will not be scheduled for publication until the required changes have been made.

Yours sincerely,

Mónica P. Colaiácovo

Academic Editor

PLOS Genetics

Gregory P. Copenhaver

Editor-in-Chief

PLOS Genetics

Comments from the reviewers (if applicable):

Reviewer's Responses to Questions

**Comments to the Authors:**

Reviewer #1: This much improved manuscript demonstrates the continued requirement for the conserved ZYG-9/TAC-1 complex in the maintenance of acentrosomal meiotic spindle poles using fast acting ts mutants. The results provide important reproducibility to a recently published Cavin-Meza 2022 paper and there are several important results not reported by Cavin-Meza.

I suggest two minor additions to the text.

Original reviewer 1 commented: "It would enhance the readers understanding of the significance of the current work if a paper was cited in which continuous function was NOT required and it was shown that the negative result was not due to incomplete depletion/inactivation"

Author's response: "we are not aware of any publications that document a

conditional null condition that could provide an example of a definitive early-only

requirement for a later defect."

In the introduction, the authors should cite the following paper from Bruce Bowerman's lab which showed reciprocal time dependence of AIR-2 and ZEN-4 in cytokinesis.

The aurora-related kinase AIR-2 recruits ZEN-4/CeMKLP1 to the mitotic spindle at metaphase and is required for cytokinesis.

Severson AF, Hamill DR, Carter JC, Schumacher J, Bowerman B.

Curr Biol. 2000 Oct 5;10(19):1162-71. doi: 10.1016/s0960-9822(00)00715-6.

PMID: 11050384

The authors should add 2 sentences to their discussion citing the Cavin-Meza 2022 ZYG-9 paper.

ZYG-9ch-TOG promotes the stability of acentrosomal poles via regulation of spindle microtubules in C. elegans oocyte meiosis.

Cavin-Meza G, Mullen TJ, Czajkowski ER, Wolff ID, Divekar NS, Finkle JD, Wignall SM.

PLoS Genet. 2022 Nov 30;18(11):e1010489. doi: 10.1371/journal.pgen.1010489.

Reviewer #2: I am very happy to recommend publication after the thorough responses provided by the authors

**Have all data underlying the figures and results presented in the manuscript been provided?**

Reviewer #1: Yes

Reviewer #2: Yes

PLOS authors have the option to publish the peer review history of their article (what does this mean?). If published, this will include your full peer review and any attached files.

Reviewer #1: No

Reviewer #2: **Yes: **Federico Pelisch

**Data Deposition**

http://datadryad.org/submit?journalID=pgenetics&manu=PGENETICS-D-22-00904R1

**Press Queries**

---

## [Editor Report · Acceptance letter]

3 Jan 2023

PGENETICS-D-22-00904R1 

C. elegans XMAP215/ZYG-9 and TACC/TAC-1 act at multiple times during oocyte meiotic spindle assembly and promote both spindle pole coalescence and stability 

Dear Dr Bowerman, 

We are pleased to inform you that your manuscript entitled "C. elegans XMAP215/ZYG-9 and TACC/TAC-1 act at multiple times during oocyte meiotic spindle assembly and promote both spindle pole coalescence and stability" has been formally accepted for publication in PLOS Genetics! Your manuscript is now with our production department and you will be notified of the publication date in due course.

With kind regards,

Bernadett Koltai

PLOS Genetics

On behalf of:
